# Discontinuous Galerkin Neural Operator for Pathology Defocus Deblurring

**Shaoqing Duan** [1]  **Haofei Song** [1]  **Xintian Mao** [1]  **Qingli Li** [1]  **Yan Wang** [1]

## Abstract

Defocus deblurring in pathological microscopy remains challenging due to the spatially varying and locally discontinuous nature of optical blur induced by a position-dependent integral imaging process. Existing deep learning methods, constrained by shift-invariance assumptions and limited interpretability, are not well suited to such heterogeneous blur patterns. Neural operators provide a principled alternative by modeling defocus formation directly as an integral operator, offering a new perspective on defocus deblurring. However, most existing neural operator architectures for low-level vision rely on globally parameterized kernels that assume smoothness and stationarity, limiting their ability to model heterogeneous and locally discontinuous blur patterns. To address this limitation, we propose the Discontinuous Galerkin Neural Operator (DGNO), which parameterizes the integral kernel using a discontinuous Galerkin formulation with element-local volume operators and interface numerical fluxes. DGNO provides a principled combination of locality, heterogeneity modeling, and global coherence while preserving the underlying physics of optical image formation. Extensive and insightful experiments demonstrate that DGNO surpasses state-of-the-arts, delivering sharper reconstructions, robust handling of spatially varying blur, and scalable high-resolution performance. The code will be released at https://github.com/DeepMed-Lab-ECNU/Single-Image-Deblur.

## 1. Introduction

Defocus deblurring for pathological microscopic images is of critical importance, as optical defocus can severely

[1]Shanghai Key Laboratory of Multidimensional Information Processing, East China Normal University, Shanghai, China. Correspondence to: Yan Wang <ywang@cee.ecnu.edu.cn>.

*Proceedings of the 43ʳᵈ International Conference on Machine Learning*, Seoul, South Korea. PMLR 306, 2026. Copyright 2026 by the author(s).

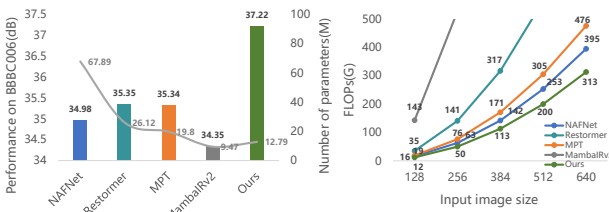

*Figure 1.* Comparisons Our DGNO and other state-of-the-art algorithms. Performance and parameters on BBBC006$_{w1}$ (left) and FLOPs (right) for defocus deblurring.

degrade cellular morphology and compromise downstream pathological analysis (Zhang et al., 2022), including cell detection (Schmidt et al., 2018) and segmentation (Keaton et al., 2023). Many image defocus deblurring networks have been proposed. However, when these methods are directly applied to pathological defocus removal, their performance is often far from satisfactory, as illustrated in Fig. 1 (see NAFNet (Chen et al., 2022), Restormer (Zamir et al., 2022), MPT (Zhang et al., 2024), and MambaIRv2 (Guo et al., 2025)). This is because most existing defocus deblurring methods aim to learn a finite-dimensional mapping $\mathcal{F} : \mathbb{R}^N \longrightarrow \mathbb{R}^N$, where both the blurred and sharp images are represented as discrete pixel vectors defined on a fixed sampling grid (Nah et al., 2017; Tao et al., 2018), without any support from physical imaging process. But in the real world, physically faithful defocus deblurring process corresponds to a function-to-function mapping $\mathcal{G} : g(\cdot) \longmapsto h(\cdot)$, which models the continuous image formation process over the spatial domain (Kovachki et al., 2023; Li et al., 2020; Lu et al., 2021). To the best of our knowledge, this perspective has rarely been exploited.

To understand the origin of pathological defocus blur, we consider the physical image formation process. When light from an object point $(\xi, \eta)$ propagates through a lens under imperfect focus, it spreads on the image plane according to a position-dependent point-spread function (PSF), as shown in Fig. 2 (a). In practice, the defocus PSF is often well-approximated by a Gaussian-like disk kernel whose scale varies with the local degree of defocus (Quan et al., 2021; 2024). As a result, the intensity at each image location $(x, y)$ is generated by an aggregation of contributions from a neighborhood region of the object domain. Classical Fourier optics (Goodman, 2005) formalizes this process through a

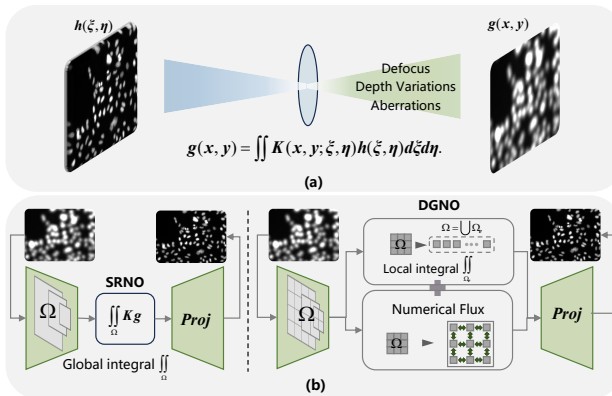

*Figure 2.* Overview of defocus-blur formation and operator-learning approaches. (a) Defocus-Blur Formation. (b) SRNO versus the proposed DGNO.

spatially varying integral operator:

$$g(x,y) = \iint K(x,y;\xi,\eta)\, h(\xi,\eta)\, d\xi d\eta, \qquad (1)$$

where $K$ denotes the PSF determined by the optical system. Only under the restrictive assumption of shift invariance does this formulation reduce to a standard convolution. But, this assumption is rarely satisfied in pathological microscopic imaging. Depth variations, heterogeneous tissue structures, refractive index inhomogeneity, and spatially varying aberrations cause the PSF to vary across the field of view, making convolution inadequate. Besides, pathological microscopic images are characterized by piecewise structural heterogeneity, where local regions remain internally stable but exhibit consistent inter-region transitions. Thus, pathology defocus deblurring should be formulated as an inverse problem of the spatially varying integral operator in Eq. (1), where the goal is to recover the latent sharp image $h$ from the observed blurred image $g$. This inverse mapping is severely ill-posed due to the position-dependent and locally discontinuous nature of the PSF.

These challenges motivate learning-based approaches that directly approximate the inverse mapping of the defocus operator from data, rather than relying on explicit kernel estimation or convolutional assumptions. Following this paradigm, a wide range of deep learning models have been applied to defocus deblurring, including convolutional neural networks (CNNs), vision transformers (ViTs), and more recent state-space models such as Mamba. CNN-based methods (Quan et al., 2021; Cho et al., 2021; Chen et al., 2022; Quan et al., 2024) implicitly assume shift-invariant convolution and therefore struggle to model spatially varying blur, while transformer-based approaches rely on global self-attention to capture long-range dependencies without physical interpretability (Liu et al., 2021; Zamir et al., 2022; Zhang et al., 2024). Recent state-space models, such as Mamba (Guo et al., 2024; 2025), further improve computational efficiency by reducing complexity to linear time;

however, they remain physically unstructured and operate at the feature-sequence level, making them insufficient for modeling the spatially varying integral operators that govern defocus blur. Despite their architectural differences, these approaches fundamentally treat defocus deblurring as a finite-dimensional image regression problem, rather than as the inversion of a spatially varying integral operator.

From this perspective, defocus deblurring is more appropriately viewed as an operator learning task, where the goal is to approximate the inverse of a spatially varying integral operator acting on **function spaces**. Recently, Neural Operators (NO) (Li et al., 2020; Kovachki et al., 2023) have emerged as a powerful framework for learning mappings between infinite-dimensional function spaces and directly parameterizing integral operators, naturally aligning with this formulation. Nevertheless, most existing NO used in low-level vision depend on global kernel parameterizations, including Diffusion Fourier Neural Operators (DiffFNO) (Liu & Tang, 2025) and Super-Resolution Neural Operator (SRNO) (Wei & Zhang, 2023). These approaches implicitly assume smoothness and stationarity, making them unsuitable for the highly localized and spatially heterogeneous behavior of real defocus blur. These motivate a defocus deblurring framework that retains the advantages of NO while explicitly incorporating locality, heterogeneity, and discontinuity awareness, which are essential for restoring images degraded by real optical blur.

To this end, we propose the **Discontinuous Galerkin Neural Operator (DGNO)**, a new operator-learning framework inspired by the discontinuous Galerkin (DG) method (Hesthaven & Warburton, 2008). As illustrated in Fig. 2(b), DGNO parameterizes the integral kernel in a DG-style manner by decomposing the global integral kernel into element-local operators and interface numerical fluxes. The former explicitly models the spatial locality inherent to real defocus blur, while the latter enables controlled cross-element information exchange at element interfaces without over-smoothing local structures. Moreover, DGNO supports both general face-based numerical flux formulations and a lightweight zero-order DG (P0DG) approximation, allowing interface coupling to be constructed either from face-wise operators or directly from element-local volume operators. By unifying element-wise operator learning with flux-based interface coupling, DGNO achieves a principled balance between local adaptability and global consistency, enabling effective modeling of spatially heterogeneous and locally discontinuous defocus blur while preserving coherent global restoration behavior beyond globally parameterized neural operators. Our contributions can be summarized as follows:

- We present the first neural operator formulation of defocus deblurring by grounding the model in the fact that defocus blur arises from a spatially varying, locally supported integral operator, making neural operators a

physically aligned alternative to other architectures.

- Building on this perspective, we propose the Discontinuous Galerkin Neural Operator (DGNO), which decomposes the global integral operator into element-local volume integral operators and interface numerical fluxes, including both a general interface-based flux formulation and a lightweight Zero-Order DG (P0DG) approximation for flexible cross-element coupling.

- Extensive experiments demonstrate that DGNO effectively captures spatially varying and discontinuous defocus blur, achieving superior restoration quality compared to state-of-the-art neural operator and image restoration approaches.

## 2. Related Work

**Defocus Deblur.** Defocus deblurring aims to restore images degraded by spatially varying defocus blur caused by optical defocusing. Conventional methods typically follow a two-stage paradigm that first estimates a defocus map (Shi et al., 2015; Karaali & Jung, 2017; Zhao et al., 2019) and then applies non-blind deconvolution using hand-crafted blur kernels (Yuan et al., 2008; Ren et al., 2018; Nan & Ji, 2020), which often suffers from inaccurate kernel modeling and ringing artifacts (Yuan et al., 2007). Recent deep learning approaches adopt end-to-end CNN-, Transformer-, or state-space-model-based architectures to handle spatially varying blur (Cho et al., 2021; Quan et al., 2021; Liu et al., 2021; Zamir et al., 2022; Zhang et al., 2024; Guo et al., 2025), but they tend to favor either local processing or global modeling, relying on implicit shift-invariance assumptions or lacking physical interpretability.

**Neural Operators.** Neural operators have recently emerged as a powerful framework for learning mappings between infinite-dimensional function spaces, enabling discretization invariant solutions of partial differential equations (Li et al., 2020; Lu et al., 2021; Kovachki et al., 2023). Fourier Neural Operators (FNOs) parameterize operators through global spectral representations, demonstrating strong capability in modeling long-range dependencies and generalizing across resolutions. In low-level vision, (Wei & Zhang, 2023) extend neural operators with kernel-based Galerkin-type attention to approximate integral operators, enabling resolution-invariant super-resolution via dynamic latent basis learning, while (Liu & Tang, 2025) enhance neural operators with spectral representations and hybrid spatial–frequency fusion mechanisms to better preserve high-frequency details in arbitrary-resolution image reconstruction. Despite these advances, most existing neural operator formulations implicitly assume global continuity of the underlying function and rely on globally coupled representations, which are suboptimal for problems characterized by spatially varying or piecewise behaviors. In contrast, our work introduces a discontinuous Galerkin neural operator that decomposes the global operator into element-local volume operators and interface fluxes, providing a structured mechanism to balance localized modeling and global coupling, and offering a new perspective for defocus deblurring.

## 3. Method

In this section, we first introduce the formulation of DGNO. Then the parameterized integral operators are presented, including the element-local volume integral operator and the interface flux operator under both the interface-based DG formulation and its P0DG approximation. Finally, we present the discrete operator assembly and network details.

### 3.1. Neural Operator Preliminaries

We consider the problem of learning mappings between function spaces. Given an input function $a : D \to \mathbb{R}^{d_a}$, a neural operator aims to learn an operator $\mathcal{G}$ such that $u(\cdot) = \mathcal{G}(a(\cdot))$. Following the neural operator framework, the input function is first lifted to a higher-dimensional latent representation by a pointwise mapping $z_0(x) = P(a(x))$, where $P$ is applied independently at each location. The latent feature field is then updated through a sequence of operator layers $z_0 \mapsto z_1 \mapsto \cdots \mapsto z_T$, and the final output is obtained via a pointwise projection $u(x) = Q(z_T(x))$. Each operator layer follows a residual form consisting of a local linear term and a non-local integral operator

$$z_{t+1}(x) = \sigma(W z_t(x) + (\mathcal{K} z_t)(x)), \qquad (2)$$

where $\sigma$ denotes a pointwise nonlinearity, $W$ is linear operator. The non-local operator $\mathcal{K}$ is defined as a kernel integral operator acting on the entire domain

$$(\mathcal{K} z)(x) = \int_D \kappa_\phi(x, y) \, z(y) \, dy, \qquad (3)$$

with $\kappa_\phi(x, y)$ denoting a learnable kernel function parameterized by $\phi$. Eq. (2) – (3) define a continuous function-to-function mapping and serve as the strong-form representation of neural operators.

### 3.2. Discontinuous Galerkin Neural Operator

**DGNO Formulation.** In discontinuous Galerkin (DG) methods for solving partial differential equations, **integration by parts decomposes divergence-form differential operators into element-local volume terms and interface surface terms**, where cross-element interactions are mediated by numerical fluxes on element boundaries (Appendix A). Inspired by this volume–interface decomposition, DGNO introduces an operator-level DG structure into the integral formulation of neural operators in Eq. (3). Specifically, the integral term in Eq. (3) is a parameterized kernel

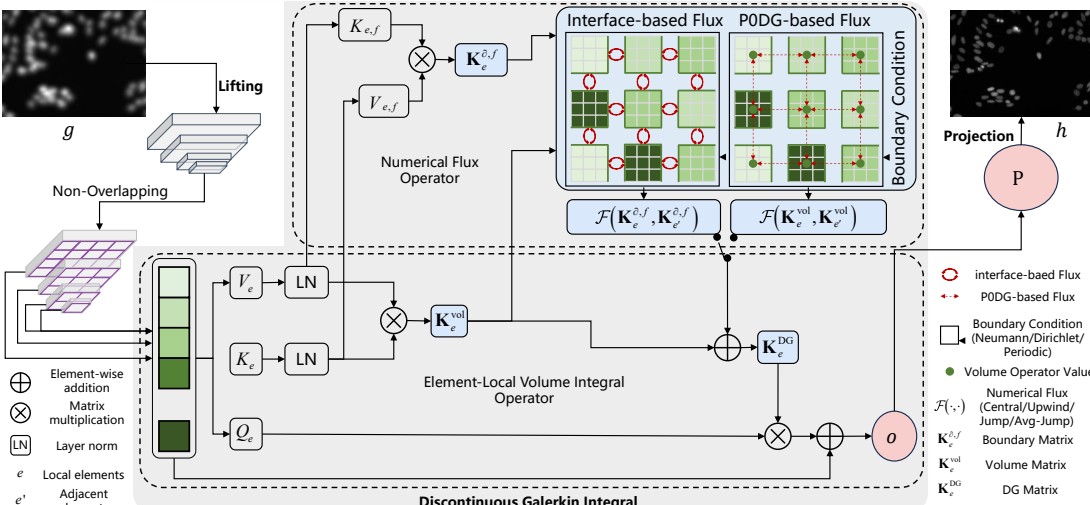

*Figure 3.* The proposed Discontinuous Galerkin Neural Operator (DGNO) architecture for defocus deblurring by lifting a defocus image $x(r)$ into a feature space using a mamba encoder. Kernel integrals composed of $T$ layers of discontinuous attention. This pipeline generates $h$, a sharp images of the input image $g$.

integral operator and does not contain differential terms, so it cannot be directly decomposed by integration by parts. Instead, DGNO decomposes the global kernel integral operator into element-wise local operators and interface coupling operators. The local integral part follows the element-wise neural operator construction, while the interface part is modeled using classical numerical fluxes.

Following the neural operator update Eq. (2), the DGNO iterative update is defined globally on the domain $D$ as

$$z_{t+1}(x) = \sigma\big(W z_t(x) + (\mathcal{K}^{\mathrm{DG}} z_t)(x)\big), \quad x \in D, \quad (4)$$

where $\mathcal{K}^{\mathrm{DG}}$ denotes the DG neural operator. Given a partition of the spatial domain into non-overlapping elements $D = \bigcup_{e=1}^{E} D_e$, the global DG operator is assembled from element-local operators as

$$(\mathcal{K}^{\mathrm{DG}} z)(x) = (\mathcal{K}_e^{\mathrm{DG}} z)(x), \quad x \in D_e, \; e = 1, \dots, E, \quad (5)$$

where $\mathcal{K}_e^{\mathrm{DG}}$ denotes the element-local DG neural operator on element $D_e$. Accordingly, the element-local DG neural operator is written as

$$(\mathcal{K}_e^{\mathrm{DG}} z)(x) = (\mathcal{K}_e^{\mathrm{vol}} z)(x) + \sum_{f \subset \partial D_e} (\mathcal{K}_{e,f}^{\mathrm{flux}} z)(x), \quad x \in D_e, \quad (6)$$

$$(\mathcal{K}_e^{\mathrm{vol}} z)(x) = \int_{D_e} \kappa_\phi(x, y)\, z(y)\, dy, \quad (7)$$

$$(\mathcal{K}_{e,f}^{\mathrm{flux}} z)(x) = \int_f \kappa_\phi(x, y)\, \widehat{\mathcal{F}}(z_e(y),\, z_{e'}(y))\, dy. \quad (8)$$

Here $(\mathcal{K}_e^{\mathrm{vol}} z)(x)$ denotes the element-local kernel volume integral on the $D_e$, and $(\mathcal{K}_{e,f}^{\mathrm{flux}} z)(x)$ represents the numerical flux operator associated with the interface $f \subset \partial D_e$ and its neighboring element. The function $\widehat{\mathcal{F}}(\cdot, \cdot)$ is a numerical

flux that combines the interface information from both sides. Specifically, $z_e(y)$ and $z_{e'}(y)$ denote the evaluations of the latent field on the interface $f$ taken from the interiors of the two adjacent elements $D_e$ and $D_{e'}$, respectively.

**Element-local Volume Integral Operator.** Following the Galerkin-type attention formulation (Wei & Zhang, 2023), the integral operator is parameterized through learned query, key, and value functions $q(x) = W_q z(x)$, $k(x) = W_k z(x)$, and $v(x) = W_v z(x) \in \mathbb{R}^d$. The element-local volume kernel integral operator Eq. (7) then can be written in a component-wise form ($j = 1, \dots, d_z$) as

$$\big((\mathcal{K}_e^{\mathrm{vol}} z)(x)\big)_j \approx \sum_{\ell=1}^{d_z} \left( \int_{D_e} k_\ell(y)\, v_j(y)\, dy \right) q_\ell(x), \quad (9)$$

which expresses the output as a linear combination of the learned basis functions $\{q_\ell(x)\}_{\ell=1}^{d_z}$, with coefficients given by element-local inner products between key and value functions. The Eq. (9) are approximated by a Monte-Carlo quadrature over samples $\{y_i^e\}_{i=1}^{n_e} \subset D_e$ as

$$\int_{D_e} k_\ell(y)\, v_j(y)\, dy \approx \frac{1}{n_e} \sum_{i=1}^{n_e} k_\ell(y_i^e)\, v_j(y_i^e), \quad (10)$$

where $n_e = p^2$ for a $p \times p$ window. Let $Q_e, K_e, V_e \in \mathbb{R}^{h \times n_e \times d}$ denote the multi-head query, key, and value evaluations on element $D_e$ where $d = C/h$, and whose column vectors represent learned basis functions spanning subspaces of their respective latent representation Hilbert spaces. The element-local volume integral operator is then obtained by applying the query representations

$$(\mathcal{K}_e^{\mathrm{vol}} z)(x) \approx Q_e \mathbf{K}_e^{\mathrm{vol}} = \frac{1}{p^2}\, Q_e\, \tilde{K}_e^\top \tilde{V}_e, \quad (11)$$

where $\tilde{K}_e = \mathrm{Ln}(K_e), \tilde{V}_e = \mathrm{Ln}(V_e)$ with $\mathrm{Ln}(\cdot)$ denoting the layer normalization and $\mathbf{K}_e^{\mathrm{vol}} \in \mathbb{R}^{h \times d \times d}$ denotes the coefficient matrix of the basis functions for the element-local volume integral operator.

**Interface-based Numerical Flux Operator.** Analogously to the element-local volume integral operator, the flux operator uses the same Monte-Carlo Galerkin discretization but integrates over element interfaces instead of the entire element. For a face $f \subset \partial D_e$ shared by neighboring elements $D_e$ and $D_{e'}$, the Eq. (8) can be approximated as

$$(\mathcal{K}_{e,f}^{\mathrm{flux}} z)(x) \approx \frac{1}{n_f} \sum_{i=1}^{n_f} \kappa_\phi(x, y_i) \,\widehat{\mathcal{F}}(z_e(y_i), \, z_{e'}(y_i)), \quad (12)$$

where $n_f$ is the number of sampled points sampled on the interface $f$. Subsequently, the multi-head key and value evaluations on face $f$ are assembled into matrices $K_{e,f}, V_{e,f} \in \mathbb{R}^{h \times n_f \times d}$. The interface coefficient matrix is

$$\mathbf{K}_e^{\partial, f} = \frac{1}{n_f} \tilde{K}_{e,f}^\top \tilde{V}_{e,f} \in \mathbb{R}^{h \times d \times d}, \quad (13)$$

with $\tilde{K}_{e,f} = \mathrm{Ln}(K_{e,f})$ and $\tilde{V}_{e,f} = \mathrm{Ln}(V_{e,f})$. An analogous definition holds for $\mathbf{K}_{e'}^{\partial, f}$ on the neighboring element $D_{e'}$ sharing the same face $f$. Assuming that elements $D_e$ and $D_{e'}$ share the same basis functions on the shared face $f$, consistent with those used in the volume integral operator, the numerical flux operator can be written as

$$(\mathcal{K}_{e,f}^{\mathrm{flux}} z)(x) \approx Q_e \mathbf{K}_{e,f}^{\mathrm{flux}} = Q_e \widehat{\mathcal{F}}\left(\mathbf{K}_e^{\partial, f}, \mathbf{K}_{e'}^{\partial, f}\right), \quad (14)$$

where $\mathbf{K}_{e,f}^{\mathrm{flux}}$ denotes the coefficient matrix of the numerical flux on element $D_e$. This implies that each local element not only captures its own internal information but also exchanges information with neighboring elements through numerical fluxes, thereby enabling global coupling.

**P0DG-based Numerical Flux Operator.** Beyond the interface-based flux construction described above, the discontinuous Galerkin neural operator also admits a Zero-order DG (P0DG) formulation, in which the numerical flux is derived directly from element-local volume integral operators. Since the latent field is represented as piecewise constant on each element and exhibits no spatial variation within the element, the element-local volume operator fully characterizes the element's contribution, enabling the numerical flux across interfaces to be computed directly from the neighboring cell-wise representations without the need for any additional interface integral operations. Consequently, the numerical flux operator in the P0DG case is given by

$$\mathbf{K}_{e,f}^{\mathrm{flux}} = \widehat{\mathcal{F}}\left(\mathbf{K}_e^{\mathrm{vol}}, \mathbf{K}_{e'}^{\mathrm{vol}}\right) \in \mathbb{R}^{h \times d \times d}, \quad (15)$$

where $\mathbf{K}_e^{\mathrm{vol}}$ and $\mathbf{K}_{e'}^{\mathrm{vol}}$ denote the coefficient matrices of the element-local volume operators on the two neighboring

elements. This P0DG formulation can be interpreted as a lowest-order approximation of the interface-based flux.

**Discrete DG Neural Operator Assembly.** Combining the element-local volume operator with all interface-based flux contributions, the latent update in Eq. (6) is given by

$$(\mathcal{K}_e^{\mathrm{DG}} z)(x) \approx Q_e \left(\mathbf{K}_e^{\mathrm{vol}} + \sum_{f \subset \partial D_e} \mathbf{K}_{e,f}^{\mathrm{flux}}\right) = Q_e \mathbf{K}_e^{\mathrm{DG}}, \quad (16)$$

where $\mathbf{K}_e^{\mathrm{DG}} \in \mathbb{R}^{h \times d \times d}$ denotes the discrete DG coefficient matrix on element $D_e$. Finally, by imposing boundary conditions and numerical fluxes at the operator level, the contributions from all elements are assembled to produce the global DG representation of the neural operator. Four representative numerical fluxes, namely central, upwind, jump, and average-jump, are considered under three types of boundary conditions, including Neumann, Dirichlet, and periodic; implementation details are provided in Appendix C and Appendix D. In this paper, DGNO with an interface-based flux is referred to as **DGNO-Face**, while DGNO with a P0DG-based flux is denoted as **DGNO-Cell**.

**Network details.** The overall network architecture is illustrated in Fig. 3. DGNO first lifts the blurred input image $x$ into a multi-scale feature space, producing feature maps with channel dimensions $d_e = 48, 96, 192, 384$ across four scales. The encoder captures shared basis functions from the training distribution, while the proposed discontinuous Galerkin-type attention layers further enable instance-specific basis refinement. We adopt the multi-head attention mechanism (Vaswani et al., 2017) by partitioning the queries, keys, and values into $n_{\mathrm{heads}}$ independent heads, each with dimensionality $d_z/n_{\mathrm{heads}}$. In our implementation, we set $d_z = 48, 96, 192$ and $n_{\mathrm{heads}} = 16$, resulting in 3-, 6-, and 12-dimensional features per head, respectively. The highest-scale features (384 channels) are upsampled and fused with the 192-channel features before being passed to the integral operator. We employ only two iterations ($T = 2$) of the kernel integral operator, which already outperforms prior methods while maintaining high computational efficiency.

## 4. Experiments

To demonstrate the effectiveness of our model, following MPT (Zhang et al., 2024), we evaluate DGNO on three microscopic defocus blur datasets: $\mathrm{BBBC006}_{w1}$ (Ljosa et al., 2012), $\mathrm{BBBC006}_{w2}$ (Ljosa et al., 2012), 3DHistech (Geng et al., 2022). We provide more details of the used datasets, training settings, and additional visual results in Appendix.

**Implementation details.** The DGNO model adopts a multi-scale neural-operator architecture with four levels of lifting and three output scales. The number of encoder modules (Guo et al., 2025) from first to fourth level is $[2, 4, 6, 2]$ with

*Table 1.* Comparisons with other Single Image Defocus Deblurring methods on BBBC006$_{w1}$ (Ljosa et al., 2012), BBBC006$_{w2}$ (Ljosa et al., 2012) and 3DHistech (Geng et al., 2022).

| Method | BBBC006$_{w1}$ | | | BBBC006$_{w2}$ | | | 3DHistech | | |
|---|---|---|---|---|---|---|---|---|---|
| | PSNR↑ | SSIM↑ | LPIPS↓ | PSNR↑ | SSIM↑ | LPIPS↓ | PSNR↑ | SSIM↑ | LPIPS↓ |
| GKMNet (Quan et al., 2021) | 34.42 | 0.941 | 0.132 | 26.87 | 0.785 | 0.396 | 33.42 | 0.852 | 0.130 |
| MIMO-Unet (Cho et al., 2021) | 35.15 | 0.948 | 0.117 | 29.70 | 0.828 | 0.354 | 32.40 | 0.837 | 0.169 |
| NAFNet (Chen et al., 2022) | 34.98 | 0.947 | 0.111 | 29.44 | 0.825 | 0.341 | 33.23 | 0.889 | 0.132 |
| SwinIR (Liu et al., 2021) | 34.75 | 0.943 | 0.123 | 30.02 | 0.829 | 0.351 | 32.57 | 0.841 | 0.136 |
| MambaIRv2 (Guo et al., 2025) | 34.35 | 0.941 | 0.113 | 30.98 | 0.823 | 0.361 | 33.48 | 0.881 | 0.106 |
| Restormer (Zamir et al., 2022) | 35.35 | 0.950 | **0.103** | 31.70 | 0.842 | 0.331 | 33.46 | 0.880 | 0.125 |
| MPT+EFCR (Zhang et al., 2024) | 35.44 | 0.947 | 0.114 | 30.48 | 0.830 | 0.348 | 33.58 | 0.887 | 0.119 |
| DGNO-Face | 37.09 | 0.958 | 0.104 | **32.66** | 0.847 | 0.323 | **34.02** | **0.890** | 0.095 |
| DGNO-Cell | **37.22** | **0.959** | 0.103 | 32.54 | **0.848** | **0.322** | 34.00 | **0.890** | **0.093** |

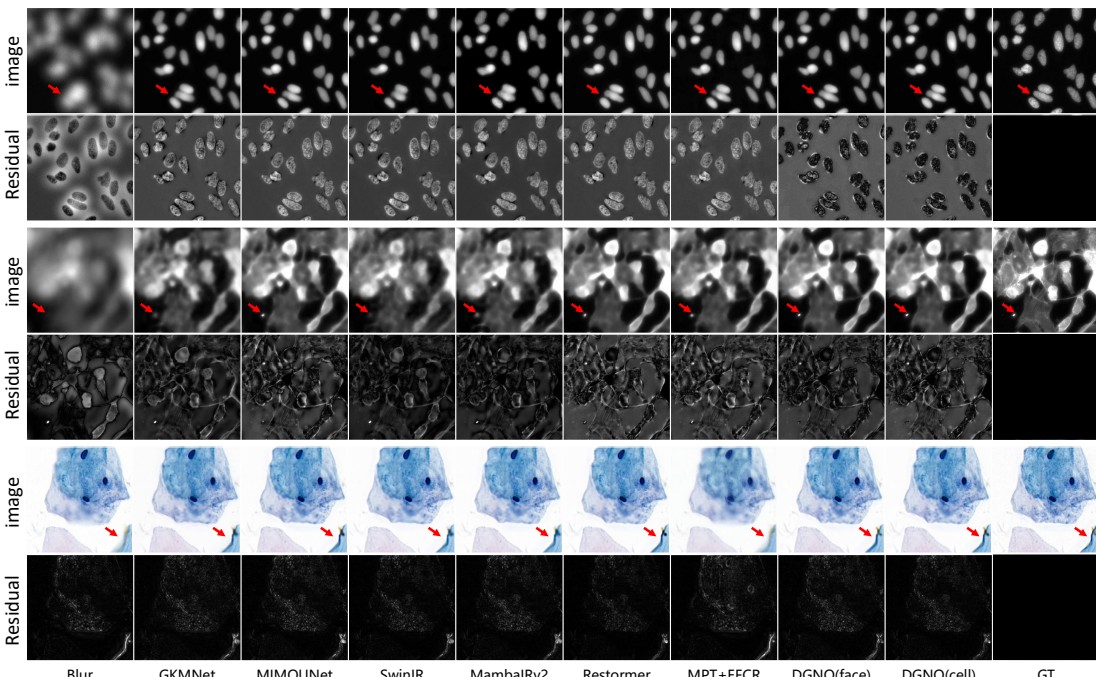

*Figure 4.* Visual comparison of single image defocus deblur approaches on BBBC006$_{w1}$ and BBBC006$_{w2}$.

channel dimensions [48, 96, 192, 384]. DGNO operates on non-overlapping local elements of size $8 \times 8$. The model was trained on NVIDIA RTX4090 GPUs (48 GB) using AdamW optimizer ($\beta_1 = 0.9$, $\beta_2 = 0.999$, weight decay=$1 \times 10^{-4}$) with an initial learning rate of $3 \times 10^{-4}$ (cosine decay to $1 \times 10^{-6}$). The batch size is 8 with training patches in the size of $256 \times 256$ image patches with data augmentation including horizontal flipping and vertical flipping.

### 4.1. Comparison with State-of-the-Art

We compare the proposed DGNO with representative state-of-the-art single-image defocus deblurring methods on BBBC006 (Ljosa et al., 2012) and 3DHistech (Geng et al., 2022), as summarized in Table 1. On BBBC006$_{w1}$, DGNO achieves the best performance, reaching 37.22/37.09 dB with 50 FLOPs and 12.79M parameters, exceeding Restormer and MPT+EFCR by up to 1.87 dB. On the more challenging BBBC006$_{w2}$ dataset, DGNO further outper-

forms MPT and Restormer by 2.18 dB and 0.96 dB, respectively. On the 3DHistech dataset, both DGNO-Face and DGNO-Cell yield similar PSNR results, and both outperform MPT+EFCR by 0.44 dB. Note that DGNO-Cell outperforms DGNO-Face on BBBC006$w1$, whereas the opposite trend is observed on the more challenging BBBC006$w2$ dataset. This observation suggests that DGNO-Cell, owing to its piecewise-constant approximation, is better suited to relatively simpler imaging scenarios, while DGNO-Face is more effective in handling more complex structural patterns. More discussions regarding stability and generalizability on DGNO-Cell and DGNO-Face are discussed in Sec. 4.2.

### 4.2. Analysis and Discussion

This section analyzes the proposed DGNO framework by comparing discontinuous Galerkin operators with global Galerkin operators, examining numerical flux and boundary condition designs, discussing generalization to real-world

*Table 2.* Comparison with the global Galerkin operator on BBBC006$_{w1}$ (Ljosa et al., 2012).

| METHOD | BBBC006$_{w1}$ PSNR | SSIM | LPIPS | PARAMS (MB) | FLOPs (G) |
|---|---|---|---|---|---|
| SRNO(GG) | 36.71 | 0.957 | 0.104 | 12.79 | 50.12 |
| SRNO+WIN(LG) | 36.85 | 0.957 | 0.104 | 12.79 | 50.12 |
| DGNO-FACE | 37.07 | 0.958 | 0.104 | 12.79 | 50.15 |
| DGNO-CELL | **37.21** | **0.959** | **0.103** | 12.79 | 50.12 |

defocus and evaluating generalization across lifting modules, defocus levels, and downstream tasks.

**From Global Galerkin Operators to Discontinuous Galerkin Operators.** Table 2 presents an ablation study comparing the proposed DGNO with a global Galerkin (GG) operator (SRNO) and its local windowed variant on BBBC006$_{w1}$. Although GG was initially developed for image super-resolution (Wei & Zhang, 2023), aiming at learning global smoothness and arbitrary upsampling, we are the first to explore its performance in pathological image defocus deblurring from the physical imaging process perspective. As a strong baseline, the GG operator achieves a PSNR of 36.71 dB, outperforming the Transformer-based MPT (35.44 dB), thereby highlighting the advantage of neural operator–based modeling for defocus deblurring. Building upon this baseline, introducing a window-based local Galerkin (LG) operator (SRNO+Win) yields a PSNR gain of 0.14 dB, indicating the benefit of enforcing locality consistent with the local integral nature of defocus blur. Going beyond windowed locality, DGNO with interface-based numerical flux further improves the PSNR to 37.07 dB without increasing the computational complexity. Finally, adopting the P0DG-based flux achieves the best performance across all metrics, demonstrating that DG operators provide a more faithful operator-level modeling of defocus blur.

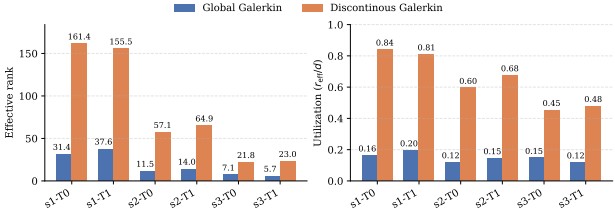

*Figure 5.* Effective-rank analysis of latent representations across scales and iteration steps ($T = 0, 1$). (a) the absolute effective rank of the latent matrix $z$, (b) the effective-rank utilization $r_{\text{eff}}/d$

To validate the discontinuous pattern learning ability of DGNO, as shown in Fig. 5, we evaluate the representation capacity of GG and DG operator via the effective rank of the latent matrix $z$ and its utilization. **Note that the higher the rank, the more discontinuous patterns the model can learn.** Across all scales ($s1$: $H/4 \times W/4$, 192-dim; $s2$: $H/2 \times W/2$, 96-dim; $s3$: $H \times W$, 48-dim) and iteration steps ($T = 0, 1$), DGNO-Face consistently achieves higher effective rank and utilization than the GG operator. These results demonstrate that DGNO enriches basis diversity and

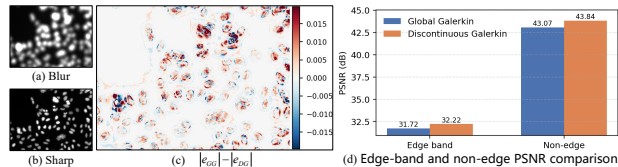

*Figure 6.* Residual visualization and edge-aware quantitative analysis comparing GG and DG operators. (a) Blur image (b) Sharp image. (c) Absolute error difference $|e_{\text{GG}}| - |e_{\text{DG}}|$. (d) PSNR comparison on edge-band and non-edge regions.

*Table 3.* Ablation on Different Flux and Boundary Condition. Neumann Boundary (NB), Dirichlet Boundary (DB) and Periodic Boundary (PB).

| FLUX | DGNO-FACE NB | DB | PB | DGNO-CELL NB | DB | PB |
|---|---|---|---|---|---|---|
| CENTRAL | 36.98 | 36.99 | 36.86 | 36.91 | 37.06 | 36.88 |
| JUMP | 37.02 | 37.06 | 37.06 | **37.22** | 37.16 | 36.97 |
| AVG-JUMP | 37.02 | **37.09** | 36.99 | 37.02 | 36.85 | 37.01 |
| UPWIND | 36.95 | 37.03 | 36.88 | 37.08 | 37.03 | 36.92 |

prevent rank collapse during iterative operator refinement.

To further analyze the boundary-aware behavior of the DG operator, Fig. 6 presents a qualitative and quantitative comparison with the GG operator on the BBBC006$_{w1}$ dataset. The error difference map in Fig. 6(c), defined as $|e_{\text{GG}}| - |e_{\text{DG}}|$, where $|e_{\text{GG}}|$ and $|e_{\text{DG}}|$ denote the absolute reconstruction errors of the GG and DG operators, respectively, shows that DG operator consistently reduces reconstruction errors near object boundaries and fine cellular structures, while maintaining comparable accuracy in homogeneous regions. The region-wise PSNR analysis in Fig. 6(d) further confirms this trend. By separately evaluating edge-band and interior regions, DG operator achieves a PSNR improvement of $+0.49$ dB on boundary regions and also a gain of $+0.77$ dB within non-edge regions.

To analyze the interpretability and qualitative behavior of different defocus deblurring models, Fig. 7 presents a visual comparison of local attribution maps (LAM) (Gu & Dong, 2021) on the BBBC006$_{w1}$ dataset. The LAM provides spatial interpretability by revealing how each model distributes its responses across image regions. Compared with CNN- and Transformer-based baselines such as NAFNet and MPT, as well as the global Galerkin operator SRNO, the proposed DGNO exhibits more localized and boundary-aligned activations. Specifically, NAFNet exhibits activations that are distributed within a large region, which may correlate with its convolutional inductive bias, and is inconsistent with the physical image formation process. MPT shows sparse yet spatially diffuse responses, such as the activations in the upper-left region (see the blue arrow), which may correlate with the global learning ability of self-attention. In contrast, globally coupled models SRNO exhibit spatially diffuse activations indicative of excessive long-range mixing and boundary interference, as highlighted by the red arrows in Fig. 7. Correspondingly, the more localized and boundary-

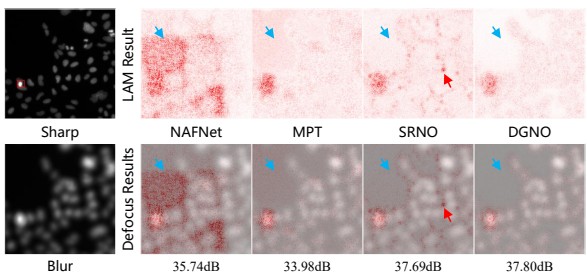

*Figure 7.* LAM-based interpretability and defocus deblurring results on BBBC006.

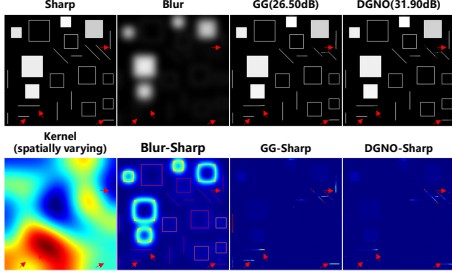

*Figure 8.* Visual comparison on a controlled synthetic example with spatially varying Gaussian blur.

*Table 4.* Quantitative comparison on BBBC006$_{w1}$ (Ljosa et al., 2012) with Seidel-coefficient-based spatially varying blur.

| Model | Blind | Uses true spatially varying PSF | PSNR | SSIM |
|---|---|---|---|---|
| Blur | - | - | 31.07 | 0.873 |
| Deconvolution | × | ×(single center PSF only) | 32.87 | 0.920 |
| Ring Deconvolution | × | ✓ | 40.15 | 0.966 |
| Ours | ✓ | × | 36.51 | 0.937 |

aligned activations produced by DGNO (highlighted by the blue arrows in Fig. 7) lead to sharper reconstruction of cellular boundaries and finer internal structures, resulting in the highest PSNR among all methods. These results show the discontinuous Galerkin formulation effectively enables controlled cross-region interactions via interface fluxes while preserving element-wise locality, leading to superior boundary fidelity and overall reconstruction quality.

**Controlled Synthetic Experiments under Spatially Varying Blur.** To more directly validate the advantage of DGNO under spatially varying blur, we conduct two controlled synthetic experiments. First, on 50 randomly generated synthetic images, we constructed sharp patterns composed of filled squares, hollow boxes, and thin lines, and applied known spatially varying Gaussian blur, where the blur strength at each pixel was controlled by a predefined sigma map. This setting isolates restoration performance under spatially heterogeneous blur with fully known degradation kernels, without interference from real-image distribution bias. DGNO consistently outperforms the global Galerkin baseline under different blur ranges: when the sigma range is 8–10, the average PSNR of GG, DGNO-Face, and DGNO-Cell is 34.69, 38.96, and 37.91 dB, respectively; when the range is expanded to 0.6–13, the corresponding values are 44.24, 46.23, and 46.32 dB. As shown in Fig. 8, GG leaves more residual blur and loses local structures around thin lines, hollow-box boundaries, and other high-frequency regions, whereas DGNO restores sharper edges and exhibits weaker residual errors. Second, on BBBC006$_{w1}$ (153 images), we further synthesized spatially varying blur using known Gaussian kernels with random spatial variation and a sigma range of 0.6–13. Under this setting, DG improves PSNR over GG by 1.94 dB (25.81 to 27.75 dB), and still achieves a 1.77 dB gain on boundary regions, indicating stronger modeling ability in spatial transition areas.

**Comparison with Non-blind Spatially Varying Deconvolution.** To evaluate performance under more realistic spatially varying blur, we compare our method on BBBC006$_{w1}$ with ring deconvolution (Kohli et al., 2025), a non-blind spatially varying deconvolution method. The spatially varying PSF is generated from Seidel coefficients estimated from a

calibration image using rdmpy, and blurred observations are synthesized via the spatially varying forward model. Table 4 compares our method with the blurred input, a non-blind deconvolution baseline using only a single center PSF, and Ring deconvolution. Our method achieves 36.51 dB / 0.937 SSIM, outperforming both the blurred input (31.07 dB / 0.873) and the single-PSF baseline (32.87 dB / 0.920). Ring deconvolution performs best (40.15 dB / 0.966), serving as a non-blind upper bound with access to the ground-truth spatially varying PSF. This result validates the effectiveness of our method in handling spatially varying blur without explicit PSF access. As shown in Fig. 9, the blurred image exhibits severe spatially varying degradation, while single-PSF deconvolution only provides limited recovery and introduces noticeable artifacts in the zoomed region. Ring deconvolution restores sharper structures with access to the true spatially varying PSF, whereas our blind DGNO recovers nuclei boundaries and fine structures.

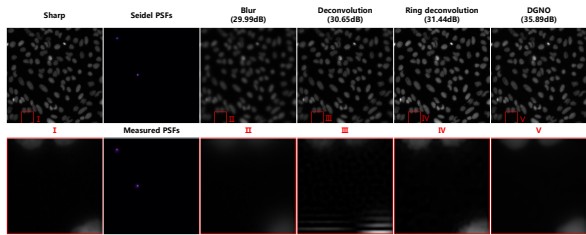

*Figure 9.* Visual comparison with non-blind spatially varying deconvolution on BBBC006$_{w1}$ under Seidel-coefficient-based blur.

**Numerical Flux and Boundary Condition Analysis.** Table 3 presents an ablation study of numerical fluxes and boundary conditions for DGNO-Face and DGNO-Cell on BBBC006$_{w1}$. For DGNO-Face, the average-jump flux under Dirichlet boundary conditions achieves the highest PSNR, whereas for DGNO-Cell, the jump flux under Neumann boundary conditions yields the best performance. Across different flux and boundary settings, DGNO-Face

exhibits more stable performance, while DGNO-Cell shows larger variability and higher sensitivity to boundary conditions. Table 12 reports an ablation on the penalty coefficient for both interface-based and cell-based numerical fluxes under their optimal settings. A learnable penalty consistently achieves the best PSNR, demonstrating the effectiveness of adaptively controlling operator-level flux strength.

**Generalization to Real-World Defocus.** We further evaluate DGNO on real-world DPDD (Abuolaim & Brown, 2020). As shown in Table 5, DGNO-Face achieves the best PSNR and competitive perceptual quality among the compared methods, with 26.42 dB PSNR and 0.173 LPIPS. The result demonstrates that the proposed operator-level formulation is not limited to pathology images and can generalize to real-world defocus blur in natural scenes.

*Table 5.* Quantitative real-world deblur evaluation on DPDD (Abuolaim & Brown, 2020).

| Model | PSNR | SSIM | LPIPS |
|---|---|---|---|
| GKMNet (Chen et al., 2022) | 25.47 | 0.789 | 0.219 |
| Restormer (Zamir et al., 2022) | 25.98 | 0.811 | 0.178 |
| MambaIR (Guo et al., 2024) | 26.11 | 0.809 | 0.202 |
| MPT (Zhang et al., 2024) | 26.21 | 0.826 | 0.175 |
| FSNet (Cui et al., 2023) | 26.22 | 0.811 | 0.207 |
| ConvIR (Cui et al., 2024) | 26.36 | 0.820 | 0.174 |
| DGNO-Face | 26.42 | 0.814 | 0.173 |

**Generality across Lifting Modules for Neural Operators.** Table 6 evaluates the generality of DGNO by instantiating the lifting operator with different backbone encoders. Across all lifting module choices, DGNO consistently yields performance improvements.

*Table 6.* Ablation with Different Lifting Modules.

| Model | PSNR | SSIM | LPIPS |
|---|---|---|---|
| NAFNet (Chen et al., 2022) | 34.98 | 0.947 | 0.111 |
| NAFNet+DGNO | **35.72** | **0.953** | **0.989** |
| Restormer (Zamir et al., 2022) | 35.35 | 0.950 | 0.103 |
| Restormer+DGNO | **35.53** | **0.952** | **0.102** |
| MPT (Zhang et al., 2024) | 35.44 | 0.947 | 0.114 |
| MPT+DGNO | **35.80** | 0.948 | **0.110** |

**Performance under Different Defocus Levels.** Table 7 reports a quantitative comparison of different algorithms under decreasing defocus blur levels ($z = 01, 05, 13$) on BBBC006$_{w1}$, where $z = 16$ corresponds to the in-focus plane. As the defocus severity decreases, all methods benefit from reduced blur, leading to consistent improvements in PSNR. DGNO-Face consistently achieves the highest PSNR across all defocus levels, demonstrating strong robustness to varying blur conditions. DGNO-Cell delivers competitive performance and outperforms most baselines; however, under stronger defocus, its effectiveness is constrained by the P0DG, which lacks explicit interface modeling and loses fine boundary information compared with DGNO-Face. Overall, these results indicate that by learning the inverse integral kernel through a neural operator, DGNO enables interpretable defocus deblurring with strong generalization and stability.The corresponding results on BBBC006$_{w2}$ are provided in Table 13 in the appendix.

*Table 7.* Generalization performance of different algorithms under decreasing levels of defocus blur ($z = 01, 05, 13$) on BBBC006$_{w1}$ (Ljosa et al., 2012), where $z = 16$ corresponds to the in-focus plane.

| | z=01 | z=05 | z=13 |
|---|---|---|---|
| GKMNet (Quan et al., 2021) | 31.50 | 33.43 | 32.78 |
| NAFNet (Chen et al., 2022) | 32.37 | 34.43 | 36.33 |
| MambaIRv2 (Guo et al., 2025) | 30.63 | 33.25 | 36.29 |
| SwinIR (Liu et al., 2021) | 30.32 | 32.66 | 34.21 |
| Restormer (Zamir et al., 2022) | 32.18 | 33.79 | 36.33 |
| MPT (Zhang et al., 2024) | 32.87 | 34.72 | 35.56 |
| DGNO-Cell | 32.41 | 34.87 | 36.25 |
| DGNO-Face | **34.30** | **36.32** | **36.99** |

*Table 8.* Cell detection results on deblurred BBBC006.

| IoU | 0.5 | 0.7 | 0.9 | Mean AP |
|---|---|---|---|---|
| blur | 0.5672 | 0.3084 | 0.0708 | 0.3154 |
| MambaIRv2 (Guo et al., 2025) | 0.7972 | 0.6786 | 0.0939 | 0.5232 |
| SwinIR (Liu et al., 2021) | 0.7914 | 0.6722 | 0.0643 | 0.5160 |
| Restormer (Zamir et al., 2022) | 0.7910 | 0.6950 | 0.1122 | 0.5287 |
| MPT+EFCR (Zhang et al., 2024) | 0.7978 | 0.7026 | 0.1167 | 0.5350 |
| DGNO-Face | 0.8055 | **0.7211** | 0.1355 | 0.5540 |
| DGNO-Cell | **0.8070** | 0.7195 | **0.1418** | **0.5561** |
| sharp | 0.8012 | 0.7175 | 0.2089 | 0.5758 |

**Validation on Downstream Tasks.** Defocus blur can severely degrade cell detection performance (Schmidt et al., 2018), which is critical for many downstream biological analyses. To evaluate the impact of defocus deblurring on cell detection, we apply StarDist (Schmidt et al., 2018) to the BBBC006 dataset before and after restoration. The results are reported in Table 8 in terms of Average Precision (AP) under different Intersection-over-Union (IoU) thresholds, where higher AP indicates more accurately detected cells. Compared with the blurred input, DGNO-based deblurring leads to substantial improvements in cell detection performance across all IoU thresholds. In particular, DGNO-cell achieves the highest mean AP of 0.5561, while DGNO-Face attains the best performance at IoU threshold of 0.7211. These results demonstrate that the proposed DGNO not only improves restoration quality but also better preserves cellular shape and boundary structures, thereby significantly benefiting downstream cell detection tasks.

## 5. Conclusion

We proposed DGNO, a novel operator learning framework for pathological image defocus deblurring. By incorporating discontinuous Galerkin principles into neural operators, DGNO decomposes the global operator into element-local volume operators and interface fluxes, enabling structured local modeling and controlled cross-element interactions. We developed both face-wise and cell-wise (P0DG) formulations of DGNO. Experimental results on three microscopy defocus deblurring datasets and one natural image defocus deblurring dataset show that DGNO consistently outperforms state-of-the-art methods in terms of reconstruction accuracy and perceptual quality.

## Acknowledgements

This work was supported by the National Natural Science Foundation of China (Grant No. 62471182), Science and Technology Commission of Shanghai Municipality Basic Research Program (Grant No. 25JD1401300), Shanghai Rising-Star Program (Grant No. 24QA2702100), and the Science and Technology Commission of Shanghai Municipality (Grant No. 22DZ2229004)

## Impact Statement

This work proposes a physically interpretable neural operator framework for pathological defocus deblurring. By modeling spatially varying blur as a piecewise integral operator, the proposed method may improve image restoration quality and support downstream microscopy analysis. The approach is intended for research use and does not directly perform clinical diagnosis.

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

## A. Discontinuous Galerkin Weak Form: Volume and Interface Terms

Let $D \subset \mathbb{R}^d$ be a bounded domain and consider an operator equation

$$\mathcal{L}(u) = f \quad \text{in } D, \tag{17}$$

where $\mathcal{L}$ denotes a (possibly nonlinear) differential operator. The domain is partitioned into non-overlapping elements $\{D_e\}_{e=1}^E$ such that $D = \bigcup_{e=1}^E D_e$. In the discontinuous Galerkin (DG) setting, trial and test functions are allowed to be discontinuous across element interfaces. Multiplying Eq. (17) by a test function $\phi$ and integrating over an element $D_e$ yields the elementwise weak statement

$$\int_{D_e} \phi \, \mathcal{L}(u) \, dx = \int_{D_e} \phi \, f \, dx. \tag{18}$$

**Integration by parts and volume–surface decomposition.** For clarity, we consider operators that admit a conservative form $\mathcal{L}(u) = \nabla \cdot \mathbf{F}(u)$. The product rule for the divergence operator gives

$$\nabla \cdot (\phi \, \mathbf{F}(u)) = \nabla \phi \cdot \mathbf{F}(u) + \phi \, \nabla \cdot \mathbf{F}(u). \tag{19}$$

Integrating Eq. (19) over $D_e$ and applying the divergence theorem yields the integration-by-parts formula

$$\int_{D_e} \phi \, \nabla \cdot \mathbf{F}(u) \, dx = - \int_{D_e} \nabla \phi \cdot \mathbf{F}(u) \, dx + \int_{\partial D_e} \phi \, \mathbf{F}(u) \cdot \mathbf{n} \, ds, \tag{20}$$

where $\mathbf{n}$ denotes the outward unit normal on $\partial D_e$. Substituting (20) into (18) leads to the weak formulation

$$- \int_{D_e} \nabla \phi \cdot \mathbf{F}(u) \, dx + \int_{\partial D_e} \phi \, \mathbf{F}(u) \cdot \mathbf{n} \, ds = \int_{D_e} \phi \, f \, dx. \tag{21}$$

The first term is an element-local *volume contribution*, while the second is a *surface contribution* on $\partial D_e$.

**Numerical flux and interface term.** Since discontinuous Galerkin methods allow trial and test functions to be discontinuous across element interfaces, the physical normal flux $\mathbf{F}(u) \cdot \mathbf{n}$ is generally not uniquely defined on $\partial D_e$. Let $u^-$ and $u^+$ denote the interior and exterior traces of $u$ on an interface. To resolve this ambiguity, the physical flux is replaced by a *numerical flux* $\widehat{F}(u^-, u^+)$, which provides a consistent and stable approximation of the inter-element flux. Substituting the numerical flux into the boundary term yields the discontinuous Galerkin weak formulation given below.

**DG weak form.** The resulting discontinuous Galerkin weak formulation on each element $D_e$ consists of an element-local volume term and an interface (flux) term:

$$- \int_{D_e} \nabla \phi \cdot \mathbf{F}(u) \, dx \; + \; \int_{\partial D_e} \phi \, \widehat{F}(u^-, u^+) \, ds = \int_{D_e} \phi \, f \, dx. \tag{22}$$

The numerical flux $\widehat{F}$ constitutes the sole mechanism for inter-element coupling and plays a central role in ensuring stability and consistency of DG discretizations. Common choices of $\widehat{F}$ include the central flux, upwind-type fluxes, and interior-penalty fluxes, each leading to a different DG scheme with distinct stability and dissipation properties. The central flux is given by

$$\widehat{F}_{\text{cen}}(u^-, u^+) = \{\mathbf{F}(u)\} \cdot \mathbf{n} = \frac{1}{2}\big(\mathbf{F}(u^-) + \mathbf{F}(u^+)\big) \cdot \mathbf{n}, \tag{23}$$

and the upwind-type (Rusanov/LLF) flux

$$\widehat{F}_{\text{LLF}}(u^-, u^+) = \{\mathbf{F}(u)\} \cdot \mathbf{n} - \frac{\alpha_e}{2}[u], \qquad [u] = u^+ - u^-, \tag{24}$$

where $\alpha_e$ denotes a local upper bound on the characteristic wave speed. For diffusion operators of the form $-\nabla \cdot (\kappa \nabla u)$, the symmetric interior-penalty Galerkin (SIPG) fluxes are commonly used:

$$\widehat{u} = \{u\}, \qquad \widehat{(\kappa \nabla u) \cdot} \, \mathbf{n} = \{\kappa \nabla u\} \cdot \mathbf{n} - \tau \, [u], \tag{25}$$

where $\{\cdot\}$ denotes the arithmetic average across the interface and $\tau > 0$ is a penalty parameter controlling the strength of inter-element coupling.

## B. Relation to Classical Discontinuous Galerkin Methods

Classical DG methods are developed for the numerical discretization of PDEs in differential form. As shown in Appendix A, the DG weak formulation decomposes into an element-local volume term and an interface flux term, arising directly from integration by parts applied to the differential operator, where the numerical flux $\widehat{F}(u^-, u^+)$ resolves the non-uniqueness of the physical flux at element interfaces. DGNO, by contrast, is not derived from a differential formulation. Each operator layer in Eq. (3) is defined as a kernel integral operator over the entire domain $D$, which does not admit a natural integration-by-parts structure, so no interface terms arise at the continuous operator level.

DGNO instead introduces discontinuities at the level of operator discretization. The global kernel integral is restricted to non-overlapping elements via domain partitioning $D = \bigcup_{e=1}^{E} D_e$, yielding element-local operators as in Eq. (7). Since each local operator is a truncated restriction of the global one, the cross-element interactions encoded in the original integral are lost and must be explicitly restored through operator-valued numerical fluxes at element interfaces, constructed from the learned boundary operator matrices $\mathbf{K}_e^{\partial,f}$. The essential difference from classical DG is therefore that the volume–interface decomposition in DGNO is a deliberate discretization strategy for restoring cross-element coupling, rather than a necessary consequence of integration by parts. Despite this difference in origin, DGNO inherits the modularity, locality, and controlled global coupling of classical DG methods, and remains compatible with the neural operator learning framework.

## C. Operator-valued Numerical Fluxes.

The operator-valued numerical flux $\widehat{\mathcal{F}}(\cdot, \cdot)$ determines how interface information from neighboring elements is combined to restore inter-element coupling. We consider several standard choices inspired by classical DG discretizations, adapted to the operator-level setting.

*Central flux* is defined by averaging the boundary operators from the two sides of the interface,

$$\widehat{\mathcal{F}}_{\text{central}}\left(\mathbf{K}_e^{\partial,f}, \mathbf{K}_{e'}^{\partial,f}\right) = \frac{1}{2}\left(\mathbf{K}_e^{\partial,f} + \mathbf{K}_{e'}^{\partial,f}\right), \tag{26}$$

which yields a symmetric coupling between adjacent elements.

*Jump flux* penalizes the operator discontinuity across the interface and corresponds to a symmetric interior penalty (SIP)–style stabilization,

$$\widehat{\mathcal{F}}_{\text{jump}}\left(\mathbf{K}_e^{\partial,f}, \mathbf{K}_{e'}^{\partial,f}\right) = -\tau\left(\mathbf{K}_e^{\partial,f} - \mathbf{K}_{e'}^{\partial,f}\right), \tag{27}$$

where $\tau$ is a learnable penalty coefficient controlling the strength of inter-element coupling. Combining consistency and stabilization, the *Avg+Jump flux* takes the form

$$\widehat{\mathcal{F}}_{\text{avg+jump}}\left(\mathbf{K}_e^{\partial,f}, \mathbf{K}_{e'}^{\partial,f}\right) = \frac{1}{2}\left(\mathbf{K}_e^{\partial,f} + \mathbf{K}_{e'}^{\partial,f}\right) \quad -\tau\left(\mathbf{K}_e^{\partial,f} - \mathbf{K}_{e'}^{\partial,f}\right), \tag{28}$$

which is analogous to the classical SIP formulation and provides a balance between accuracy and stability.

Finally, we consider an *upwind-style flux* that introduces a directional bias in operator space. Specifically, we define a data-dependent weighting coefficient

$$\alpha_{e,e'}^f = \sigma\left(s\left(\mathbf{K}_e^{\partial,f}\right) - s\left(\mathbf{K}_{e'}^{\partial,f}\right)\right), \tag{29}$$

where $s(\cdot)$ denotes a scalar summary statistic of the boundary operator (e.g., the mean over matrix entries), and $\sigma(\cdot)$ is the sigmoid function. The upwind flux is then given by

$$\widehat{\mathcal{F}}_{\text{upwind}} = \alpha_{e,e'}^f \, \mathbf{K}_e^{\partial,f} + \left(1 - \alpha_{e,e'}^f\right) \mathbf{K}_{e'}^{\partial,f}, \tag{30}$$

which adaptively selects the dominant operator contribution across the interface.

## D. Boundary Conditions

Boundary conditions in the proposed DG neural operator are incorporated at the *operator level* through the operator-valued numerical flux $\widehat{\mathcal{F}}(\cdot, \cdot)$. Rather than imposing constraints directly on the latent representations or solution values, boundary

*Table 9.* Sensitivity to $W_e$ and $T$ on BBBC006$_{w1}$ (PSNR dB).

| $W_e \backslash T$ | 1 | 2 | 3 | 4 |
|---|---|---|---|---|
| 4 | 36.94 | 36.86 | 37.11 | 36.89 |
| 8 | 36.89 | **37.22** | 37.10 | 37.04 |
| 16 | 37.07 | 37.11 | 37.01 | 36.98 |
| 32 | 36.85 | 36.91 | 36.41 | 36.39 |

*Table 10.* Quantitative evaluation on RealDOF dataset (Lee et al., 2021).

| Method | PSNR | SSIM | LPIPS |
|---|---|---|---|
| DPDNet(Abuolaim & Brown, 2020) | 22.52 | 0.644 | 0.583 |
| IFAN(Lee et al., 2021) | 24.71 | 0.748 | 0.304 |
| EAMamba(Lin et al., 2025) | 24.59 | 0.761 | 0.309 |
| DGNO (Ours) | **25.08** | **0.781** | **0.275** |

*Table 11.* Memory (GB) / throughput (img/s) versus image size.

| Size | SwinIR | Restormer | MPT | DGNO |
|---|---|---|---|---|
| 128 | 4/29 | 3/24 | 2/7 | **1/30** |
| 256 | 16/5 | 12/11 | 10/9 | **4/27** |
| 384 | 36/2 | 27/5 | 24/3 | **9/18** |
| 512 | 63/1 | 48/3 | 41/2 | **15/10** |

conditions are enforced by modifying the operator-level flux evaluation on boundary faces. This treatment follows the discontinuous Galerkin philosophy and enables a unified handling of interior interfaces and physical boundaries within the same operator framework. For a boundary face $f \subset \partial D_e \cap \partial D$, the operator-valued numerical flux is evaluated by appropriately replacing the exterior operator argument according to the prescribed boundary condition.

**Dirichlet boundary conditions.** For Dirichlet boundary conditions, the exterior operator contribution is replaced by a boundary operator constructed from the given boundary data, denoted by $\mathbf{K}_{\mathrm{bc}}^{\partial,f}$. Specifically, we set $\mathbf{K}_{\mathrm{bc}}^{\partial,f} = \mathbf{K}_e^{\partial,f}$. The numerical flux on the boundary face is then evaluated as $\widehat{\mathcal{F}}\big(\mathbf{K}_e^{\partial,f}, \mathbf{K}_{\mathrm{bc}}^{\partial,f}\big) = \widehat{\mathcal{F}}\big(\mathbf{K}_e^{\partial,f}, \mathbf{K}_e^{\partial,f}\big)$.

**Neumann boundary conditions.** For Neumann boundary conditions, the prescribed normal flux on the physical boundary is incorporated directly through the operator-valued numerical flux. In the case of homogeneous Neumann conditions (zero normal flux), the exterior operator contribution is set to zero, i.e., $\mathbf{K}_{\mathrm{bc}}^{\partial,f} = \mathbf{0}$, and the numerical flux on a boundary face $f \subset \partial D_e \cap \partial D$ is evaluated as $\widehat{\mathcal{F}}\big(\mathbf{K}_e^{\partial,f}, \mathbf{0}\big)$. This corresponds to enforcing a zero-flux condition at the operator level.

**Periodic boundary conditions.** For periodic boundary conditions, boundary faces are paired and treated as interior interfaces. The boundary operators on paired faces are exchanged between the corresponding elements, and the operator-valued numerical flux is evaluated in the same manner as for interior faces. This yields a seamless coupling across periodic boundaries at the operator level.

# E. Additional Experiments and Analysis

**Sensitivity to element size and operator iterations.** We ablate DGNO-Cell over the element size $W_e$ and the number of operator iterations $T$ on BBBC006$_{w1}$. As shown in Table 9, the model is insensitive to these hyperparameters across a broad range: for $W_e = 4$–16 and $T = 1$–4, PSNR remains around 37 dB with only minor variation. The best result is obtained at $W_e = 8$ and $T = 2$. Clear degradation appears only when using a much larger element size ($W_e = 32$), especially for larger $T$, indicating that the proposed DGNO-cell is robust to moderate changes in element granularity and operator depth.

**Transfer beyond pathology deblurring.** In addition to the non-pathology DPDD evaluation in Table 5, we further evaluate DGNO on the RealDOF test set (Lee et al., 2021). Table 10 shows that DGNO achieves the best PSNR, SSIM, and LPIPS among all compared methods, further demonstrating that the proposed operator-level formulation transfers beyond pathology images. Together with the DPDD results, this supports the broader applicability of DGNO to spatially varying defocus deblurring in non-pathology domains.

**Runtime, memory, and scaling.** Complementing the performance–parameter and resolution–FLOPs analysis in Fig. 1, Table 11 reports GPU memory usage and throughput across image sizes. DGNO consistently uses substantially less memory and maintains higher throughput as the image resolution increases. This confirms that the proposed element-wise operator decomposition provides a favorable runtime–memory trade-off and scales more efficiently to larger images.

**Penalty coefficient analysis.** Table 12 evaluates the penalty coefficient in the numerical flux. The learnable coefficient achieves the best performance for both DGNO-Face and DGNO-Cell, showing that adaptive flux strength is beneficial.

**Generalization across defocus levels.** Table 13 reports generalization on BBBC006$_{w2}$ under decreasing defocus levels. DGNO remains consistently competitive across all focal planes, confirming its robustness to different blur severities.

*Table 12.* Ablation study of the penalty coefficient in the numerical flux on BBBC006$_{w1}$.

| PENALTY COEFFICIENT | DGNO-FACE PSNR | SSIM | DGNO-CELL PSNR | SSIM |
|---|---|---|---|---|
| 0.25 | 36.93 | 0.958 | 36.96 | 0.958 |
| 0.5 | 37.05 | 0.958 | 36.80 | 0.957 |
| 1 | 37.03 | 0.958 | 36.90 | 0.957 |
| LEARNABLE | **37.09** | **0.958** | **37.22** | **0.959** |

*Table 13.* Generalization performance of different algorithms under decreasing levels of defocus blur ($z = 01, 05, 13$) on BBBC006$_{w2}$ (Ljosa et al., 2012), where $z = 16$ corresponds to the in-focus plane.

| METHOD | z=01 PSNR | z=05 PSNR | z=13 PSNR |
|---|---|---|---|
| NAFNET (CHEN ET AL., 2022) | 27.45 | 29.26 | 31.46 |
| MAMBAIRv2 (GUO ET AL., 2025) | 27.43 | 29.87 | 31.37 |
| SWINIR (LIU ET AL., 2021) | 25.25 | 27.13 | 28.28 |
| RESTORMER (ZAMIR ET AL., 2022) | 27.30 | 28.75 | 32.55 |
| MPT (ZHANG ET AL., 2024) | 25.75 | 27.24 | 32.44 |
| DGNO-CELL | 29.69 | 31.54 | **32.95** |
| DGNO-FACE | **29.72** | **31.61** | 32.77 |

**Qualitative analysis of learned operators.** Figure 10 visualizes the internal operator behavior of DGNO on BBBC006 (Ljosa et al., 2012). The learned dynamic basis functions show spatially adaptive responses, indicating that DGNO captures local blur variations rather than relying on a fixed global operator. The latent representations $Q(K^\top V)$ highlight restoration-relevant cell structures and blurred boundaries. Moreover, the DG flux and boundary responses are concentrated around element interfaces and structural transitions, showing that the numerical flux effectively couples neighboring elements while preserving local spatial heterogeneity.

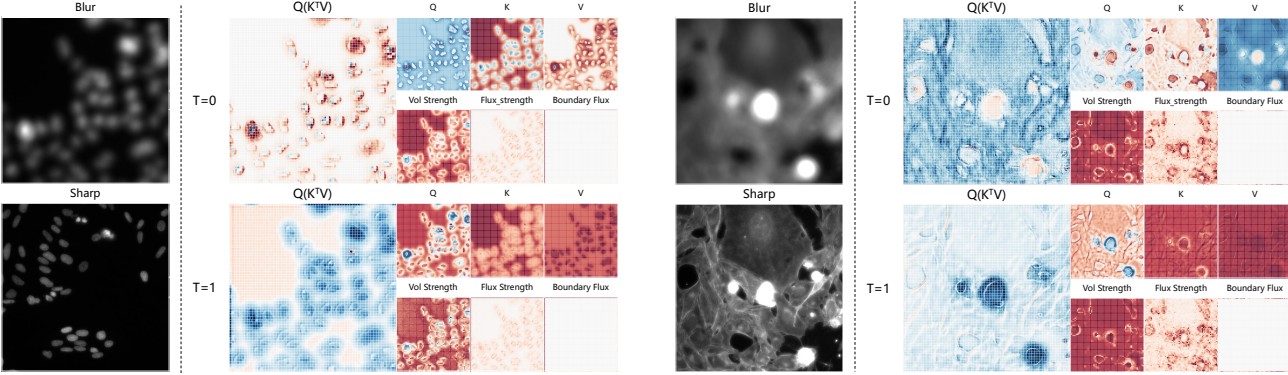

*Figure 10.* Visualization of the learned dynamic basis functions, latent representations $Q(K^\top V)$, and DG flux and boundary behavior on BBBC006 (Ljosa et al., 2012).

## F. Dataset Description.

**BBBC006** (Ljosa et al., 2012): This dataset from the Broad Bioimage Benchmark Collection (BBBC) consists of fluorescence microscopy images in two channels, denoted as $w1$ (Hoechst-stained nuclei) and $w2$ (Phalloidin-stained actin). Following the official protocol, images captured at the optimal focal plane ($z$-stack = 16) are used as sharp ground-truth references, while those at defocused planes ($z$-stack = [2, 6, 10]) serve as blurry inputs for training. All images are single-channel grayscale with a resolution of $696 \times 520$. A total of 6,144 image pairs are used, with a training/testing split of 4:1.

**3DHistech** (Geng et al., 2022): This cytopathology dataset is captured using a 3DHistech digital scanner across multiple focal planes. For each cell sample, the image containing the maximum number of cells in focus is regarded as the sharp ground truth. The dataset contains 94,973 image patches of size $256 \times 256$, divided into 66,976 for training, 9,088 for validation, and 18,909 for testing.

# G. Training Objective

To encourage restoration consistency in both spatial and frequency domains, we employ a combined multi-scale reconstruction loss:

$$L_{\text{spatial}} = \sum_{s=1}^{3} \frac{1}{E_s} \left\| \hat{Y}_s - Y_s \right\|_1 ,$$

$$L_{\text{frequency}} = \sum_{s=1}^{3} \frac{1}{E_s} \left\| \mathcal{F}(\hat{Y}_s) - \mathcal{F}(Y_s) \right\|_1 ,$$

$$L = L_{\text{spatial}} + \lambda L_{\text{frequency}}, \quad \lambda = 0.1.$$

(31)

Here, $s$ denotes the scale index, $\mathcal{F}$ is the fast Fourier transform (FFT), $E_s$ is a normalization factor, and $\hat{Y}_s$ and $Y_s$ denote the restored output and target image at scale $s$, respectively. This objective balances spatial-domain fidelity and frequency-domain consistency across scales.

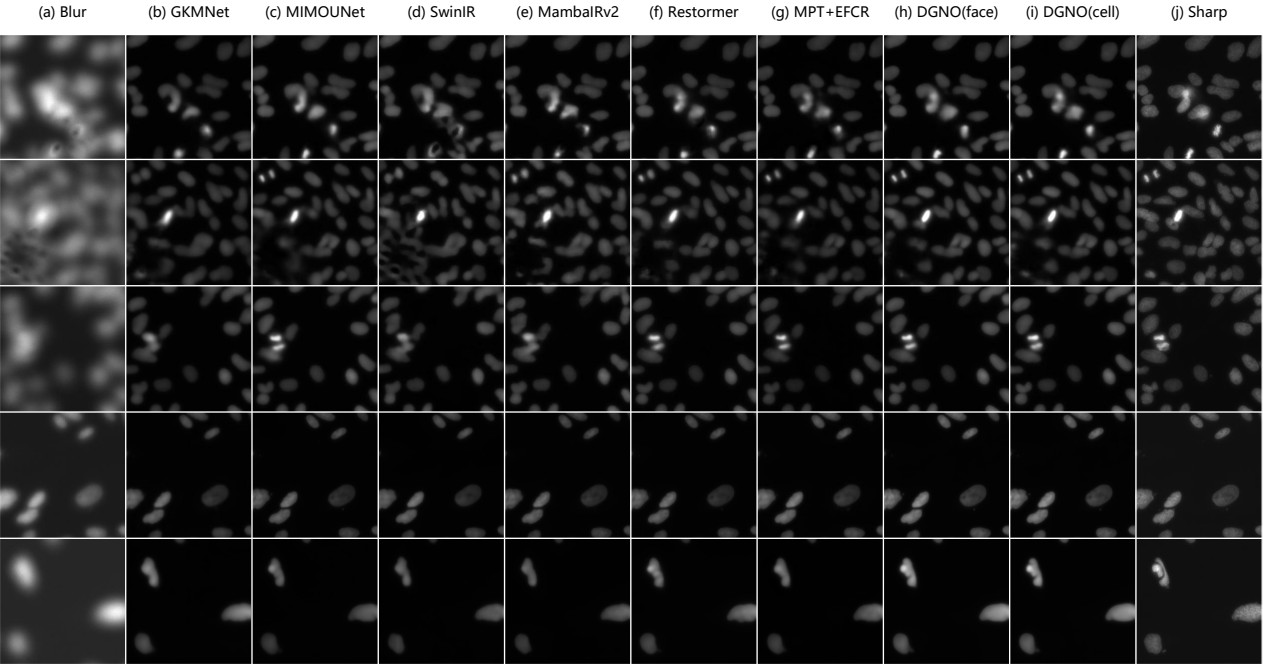

*Figure 11.* Visual comparison of single image defocus deblur approaches on BBBC006$_{w1}$ (Ljosa et al., 2012)

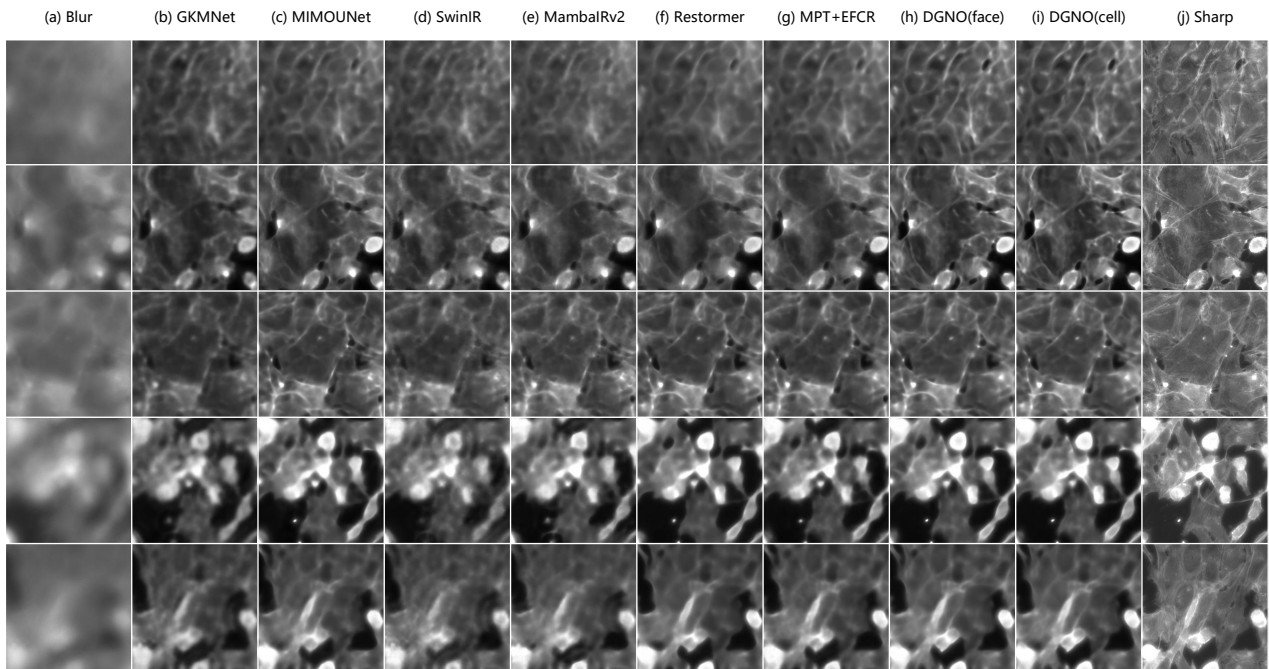

*Figure 12.* Visual comparison of single image defocus deblur approaches on BBBC006$_{w2}$ (Ljosa et al., 2012)

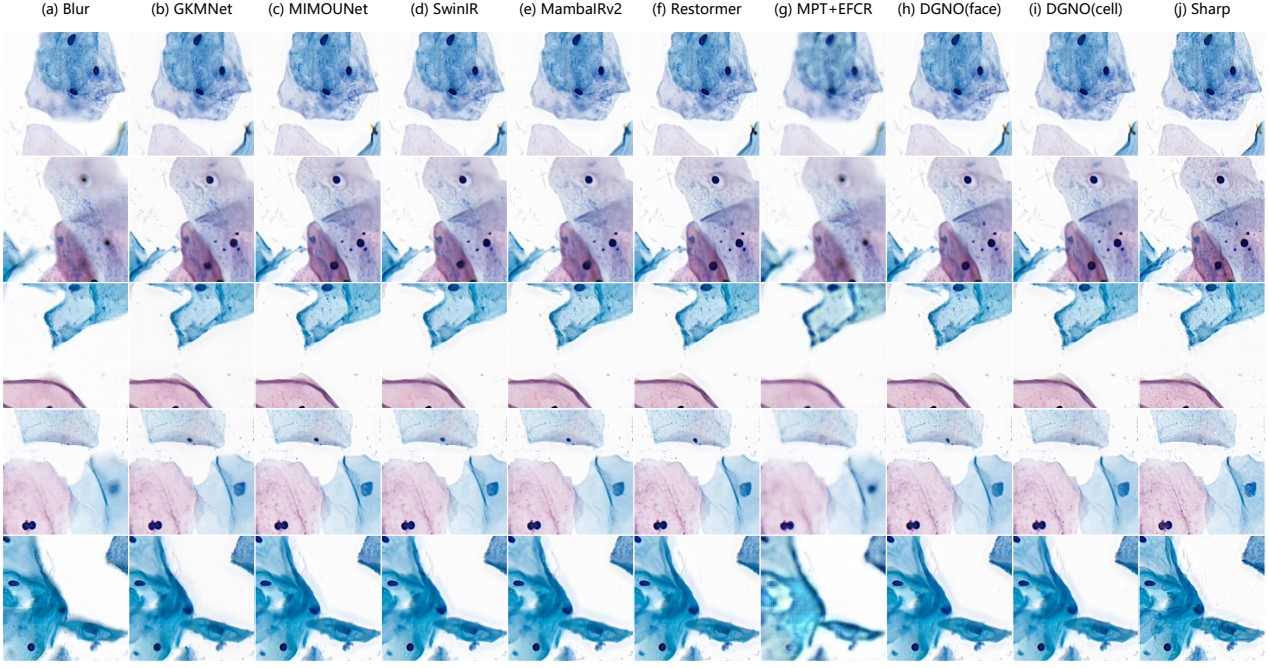

*Figure 13.* Visual comparison of single image defocus deblur approaches on 3DHistech (Geng et al., 2022)

