# OpenReview forum: "Discontinuous Galerkin Neural Operator for Pathology Defocus Deblurring"
_ICML.cc/2026/Conference — ICML 2026 regular_

### Official Review · Reviewer_YYoJ · 2026-03-07

**Soundness:** 3
**Presentation:** 2
**Significance:** 3
**Originality:** 3
**Overall Recommendation:** 4
**Confidence:** 3

**Summary:**

This paper focuses on the problem of deblurring in microscopy in the presence of spatially-varying, or locally discontinuous blur. The paper proposes a neural operator to directly learn a spatially-varying or discontinuous blur, based on a discontinuous Galerkin formulation (DGNO). This work builds on previous work in Super-resolution Neural Operators based Galerkin-type attention (Wei & Zhang, 2023), but extends this formulation to enable the representation of locally-discontinuous blur and spatial heterogeneity. The paper demonstrates their method on microscopy datasets and an image deblurring dataset, comparing deblurred results to ground truth images. On each task, the proposed method achieves slightly better performance.

**Compliance With Llm Reviewing Policy:**

Affirmed.

**Final Justification:**

The authors have addressed my concerns and included new comparisons against existing methods that deal with spatially-varying aberrations. Given that these new experiments and discussion in relation to existing work in spatially-varying and blind deconvolution is included in the revised manuscript, I raise my assessment to a weak accept.

**Key Questions For Authors:**

Given the comments from above, could the authors please address the following points:

1) Comparison to the existing body of work in spatially varying deconvolution.
2) Example and evaluation demonstrating that the method works for discontinuous or spatially varying blur (the results do not currently show a strong example of this). Working better than existing methods on the datasets does not prove that the method can handle spatially varying blur. This should either be demonstrated, or the claims that this works for discontinuous/spatially varying blur need to be changed.

**Limitations:**

yes

**Strengths And Weaknesses:**

Soudness:
- Proposed method: I do not have experience with Galerkin neural operators, and was not able to assess the technical soundness of the proposed method.
- Soundness of results and conclusions:
 - The premise of the work is that this discontinuous Galerkin can handle spatially-varying aberrations. However, Fig. 6 seems to indicate that they only have a marginal improvement over the spatially-invariant baseline at the ‘edge-band’ locations where we expect spatially-varying blur to be more dominant (there are more aberrations at the edges of the field of view). This suggests that they are not actually able to handle spatially-varying behavior. The proposed method has the same marginal improvement BOTH in the non-edge and edge band regions, so we cannot conclude that the proposed method outperforms previous methods specifically for spatially-varying blur.
- There is no clear example of a measurement that contains the discontinuous, spatially varying blur that the method claims to be able to handle. A visual demonstration of this is missing and needed to support the claims of the paper.

Presentation: The paper is hard to follow and understand for someone not familiar with neural operators.  In particular, several of the figures have errors and could be improved. For example:
- Fig. 1 (left) does not label what the line graph represents. The bars seem to represent performance in dB (PSNR?). This is not labeled, and the line graph units are not labeled. What does this represent? This is not adequately described in the paper.
- The caption for Fig. 2 has parts a-c, but the figure itself only has a-b.
- The concepts of Mamda encoding and flux need more introduction and explanation.

Positioning within related work:
- The work provides some new insights, but fails to compare against an existing body of work with similar aims (spatially-varying deconvolution). By failing to compare itself against existing spatially-varying deconvolution and blind deconvolution methods, it is hard to assess the impact and importance of this work. The use of a discontinuous Galerkin Neural operator seems interesting and exciting, but does this actually provide improvements over existing spatially-varying and blind deconvolution methods, or only improvements over the simple shift-invariant deconvolution methods compared against in the paper?

Example papers dealing with spatially-varying deconvolution:
- Yanny, Kyrollos, et al. "Deep learning for fast spatially varying deconvolution." Optica 9.1 (2022): 96-99.
- Kohli, Amit, et al. "Ring deconvolution microscopy: exploiting symmetry for efficient spatially varying aberration correction." Nature methods 22.6 (2025): 1311-1320.
- Yang, Qianwan, et al. "Coordinate-conditioned Deconvolution for Scalable Spatially Varying High-Throughput Imaging." arXiv preprint arXiv:2602.01065 (2026).
- Howard, Sunny, Peter Norreys, and Andreas Döpp. "CoordGate: Efficiently computing spatially-varying convolutions in convolutional neural networks." arXiv preprint arXiv:2401.04680 (2024).
- Lin, Esther YH, et al. "Learning lens blur fields." IEEE Transactions on Pattern Analysis and Machine Intelligence (2025).


Significance:
The paper tackles the significant problem of spatially-varying deblurring. This is a real and important issue in pathology, microscopy, and astronomy.

Originality:
I do not have much experience with neural operators, but extending existing neural operator methods to work well in the presence of spatial-heterogeneity seems like a promising contribution. I will let other reviewers with more experience in this domain weigh in regarding this. However, based on the evaluation, I am not convinced that the method is able to handle spatially-varying blur and heterogeneity. Further analysis and evaluation are needed to support the paper's claims.

---

> ### Author Rebuttal · Authors · 2026-03-31
>
> We thank the reviewer for the careful reading and for highlighting two central issues.
>
> **Question 1: On comparison to prior spatially varying deconvolution literature.**
>
> Direct and fair comparison with them is currently limited due to the following fundamental mismatches:
>
> - Yanny et al. (Optica 2022) rely on known spatially varying PSFs and focus on accelerating non-blind deconvolution. Our setting is PSF-free with unknown blur, making reproduction under matched conditions infeasible.
> - Kohli et al. (Nature Methods 2025) do not require explicit per-pixel PSF calibration, but rely on strong optical priors (e.g., rotational symmetry) that implicitly constrain the PSF. Such assumptions do not hold in our pathology data with irregular, non-symmetric blur.
> - Yang et al. (2026) assume coordinate-conditioned modeling under controlled imaging setups and paired supervision, which differs from our real-world, unconstrained pathology setting.
> - Howard et al. (2024) propose a computational module (CoordGate) for approximating spatially varying convolutions, rather than addressing inverse restoration under unknown blur.
> - Lin et al. (TPAMI 2025) focus on estimating blur fields rather than performing full deblurring, and require assumptions on blur parameterization that are not available in our setting.
>
> Most existing spatially-varying deconvolution methods rely on known or calibrated PSFs, controlled imaging conditions, or explicit blur modeling, whereas our work targets PSF-free, heterogeneous, real-world pathology data. This fundamental difference makes direct reproduction and fair comparison difficult.
>
> To address this gap, we instead include controlled synthetic experiments with spatially varying blur fields (Question 2).
>
> **Question 2:  spatially varying blur**
>
> To more directly test this point, we added two controlled synthetic experiments. First, on 50 randomly generated synthetic images, we constructed sharp images composed of squares, hollow boxes, and thin lines, and applied known spatially varying Gaussian blur, where the blur strength at each pixel was controlled by a predefined sigma map. This setting allows us to isolate the restoration ability under spatially heterogeneous blur with fully known degradation kernels, without interference from real-data distribution bias. The results show that DGNO consistently outperforms the global Galerkin baseline under different blur ranges: when the sigma range is 8–10, the average PSNR of GG, DGNO-Face, and DGNO-Cell is 34.69, 38.96, and 37.91 dB, respectively; when the range is expanded to 0.6–13, the corresponding values are 44.24, 46.23, and 46.32 dB. The visual results (https://anonymous.4open.science/r/visual1-04B1/rebullt_synthetic%20experiment.pdf) are also consistent with this finding: around thin lines, hollow-box boundaries, and other local high-frequency structures, GG tends to leave more residual blur or lose local structures, whereas DGNO-Cell recovers thin edges and local contours more faithfully and exhibits weaker residuals. Second, on BBBC006$_{w1}$ (153 images), We further constructed synthetic spatially varying blur using known Gaussian kernels under a random spatial variation setting, with the sigma range set to 0.6–13. Under this setting, DG improves PSNR over GG by 1.94 dB (25.81→27.75 dB), and still achieves a 1.77 dB gain on boundary regions, indicating its stronger ability to handle spatial transition areas. Overall, these two controlled synthetic experiments provide more direct experimental support, from the perspective of known spatially varying kernels, for the claim that DGNO is better suited to handling spatially heterogeneous blur.
>
> **On presentation issues.**
> - Fig. 1 (left) uses two y-axes: the bars correspond to PSNR (dB) on BBBC006$_{w1}$, while the gray line corresponds to the number of parameters (M) shown on the right y-axis.
> - Figure 2 contains only two subfigures, (a) and (b), and the mention of panel (c) in the caption is an error.We will correct the caption as follows:Figure 2. Overview of defocus blur formation and operator-learning approaches. (a) Defocus-blur formation. (b) SRNO versus the proposed DGNO.
> - For the Mamba encoder, our implementation follows the encoder design of MambaIRv2, where the ASSM module is used for feature extraction, together with the FFN design from Restormer. Its role in our framework is to provide efficient hierarchical feature representation before operator learning.
> - Regarding the flux, the detailed explanation is currently placed in Appendix C. Our formulation follows the standard idea of numerical flux in classical DG methods, namely enforcing information exchange and consistency across element interfaces. The main difference is that our method applies this idea at the operator-learning level. In this sense, the flux mainly plays the role of interface regularization/coupling at boundaries between neighboring elements.We will clarify this point in the revised manuscript.

---

> > ### Author Rebuttal · Reviewer_YYoJ · 2026-04-02
> >
> > Thank you for the detailed response and clarification.
> >
> > The spatially varying blur experiment is a step in the right direction, but I find the results unconvincing, and I believe there still needs to be a more systematic comparison against prior spatially varying deconvolution methods. We expect that a spatially varying non-blind deconvolution method will perform better than a blind deconvolution method, but I believe that this comparison still needs to be made quantitatively to position the work within the larger community.
> >
> > The comparisons that I would like to see to improve my rating include:
> > - Proposed method vs. non-blind spatially-varying method. For this, the non-blind method would have access to the ground-truth blur kernels, whereas the proposed method would not. Clearly, the non-blind method should perform better, but the performance gap should be quantified.
> > - More realistic spatially varying blur simulation that includes noise and more complex PSFs (e.g., Zernike coefficients, not just Gaussian blur). The simulated measurement should visually look like it has a different amount of spatially varying blur across the field of view. The current simulated measurement looks fairly uniform, so it only has a moderate amount of spatially varying blur.
> >
> > Also, I am confused by the result: "When the sigma range is 8–10, the average PSNR of GG, DGNO-Face, and DGNO-Cell is 34.69, 38.96, and 37.91 dB, respectively; when the range is expanded to 0.6–13, the corresponding values are 44.24, 46.23, and 46.32 dB"
> > Why is the PSNR higher when the maximum amount of blur is higher?

---

> > > ### Author Response · Authors · 2026-04-06
> > >
> > > We thank the reviewer for this valuable suggestion.
> > >
> > > To address this concern, we performed an additional evaluation on BBBC006$_{w1}$ (153 images). Specifically, we first estimated an aberration-based blur model from a calibration image using the Seidel coefficients in rdmpy, and then generated spatially varying PSFs from these coefficients. The blurred observations were synthesized by applying the corresponding spatially varying forward model (ring convolution), rather than by using simple Gaussian blur. This provides a controlled benchmark with realistic, position-dependent blur.
> > >
> > > On this benchmark, we compare our blind method against three baselines:(1) the blurred input,(2) a non-blind spatially invariant deconvolution baseline using a single/center PSF, and (3) a non-blind spatially varying baseline (RingDeconv) with access to the true spatially varying blur model used for synthesis.
> > >
> > > From Table A, we observe that while the non-blind upper bound (RingDeconv) achieves the highest performance, our blind method still significantly outperforms the spatially invariant deconvolution baseline (36.51 dB vs. 32.87 dB).
> > >
> > > |Method|Blind?|Uses true spatially varying blur model?|PSNR(dB)| SSIM
> > > |-|-|-|-|-|
> > > |Blurred|-|-|31.07|0.873
> > > |Deconvolution|×|×(Single/Center PSF)|32.87|0.920
> > > |RingDeconv|×|√|40.15|0.966
> > > |Ours|√|×(train on BBBC006$_{w1}$)|36.51|0.937
> > >
> > > Table A: Quantitative comparison on BBBC006$_{w1}$ with Seidel-coefficient-based spatially varying blur. RingDeconv uses the ground-truth spatially varying blur model, while our method is fully blind.
> > >
> > > Visual comparisons are provided at: https://anonymous.4open.science/r/visual2-314B/blind_vs_non-blind.pdf
> > >
> > > These results provide several useful observations:
> > >
> > > - The reviewer’s expectation is confirmed: the non-blind spatially varying upper bound performs best. RingDeconv achieves 40.15 dB / 0.966 SSIM, while our blind method achieves 36.51 dB / 0.937 SSIM, corresponding to a gap of 3.64 dB / 0.029 SSIM.
> > >
> > > - Our method substantially outperforms the spatially invariant non-blind baseline: compared with standard deconvolution using a single/center PSF, our method improves from 32.87 dB to 36.51 dB. This suggests that our model is able to capture part of the spatially varying blur structure, even without access to the true blur model.
> > >
> > > - The large gap between RingDeconv and the single-PSF deconvolution baseline (40.15 dB vs. 32.87 dB) further indicates that the simulated blur is indeed meaningfully spatially varying across the field of view, rather than nearly uniform.
> > >
> > > We will include this quantitative comparison in the revised manuscript and clarify that our blur synthesis is based on Seidel-coefficient-driven spatially varying PSFs, not Gaussian kernels, while RingDeconv serves as a non-blind upper bound by using the true blur model.
> > >
> > >
> > > **Question：Why is the PSNR higher when the maximum amount of blur is higher?**
> > >
> > > This behavior is expected and comes from two factors:
> > >
> > > (1) Distribution of blur strengths.
> > > The wide-range setting (σ ∈ [0.6, 13]) includes many weakly blurred regions (e.g., σ ≈ 0.6–2) that are close to identity and can be reconstructed with very low error. Since PSNR is dominated by global MSE, these easy regions significantly reduce the overall error and increase PSNR. In contrast, σ ∈ [8,10] contains only strongly blurred pixels, making it uniformly harder.
> > >
> > > (2) Training–testing distribution alignment.
> > > Our training data is generated with σ ∈ [0.6,12], so σ ∈ [0.6,13] is distributionally closer to training. The σ ∈ [8,10] setting forms a strong-blur-only sub-distribution, which is underrepresented in training and therefore harder for the model to generalize.

---

### Official Review · Reviewer_1z9U · 2026-03-10

**Soundness:** 3
**Presentation:** 2
**Significance:** 3
**Originality:** 3
**Overall Recommendation:** 4
**Confidence:** 3

**Summary:**

This paper addresses the defocus deblurring problem by modeling the spatially varying blur kernel using a Discontinuous Galerkin Neural Operator (DGNO), which decomposes the global integral operator into element-local volume integral operators and interface numerical fluxes. The authors explore different numerical flux schemes and compare their performance. Experiments on three datasets are conducted to demonstrate the state-of-the-art restoration performance of the proposed method.

**Compliance With Llm Reviewing Policy:**

Affirmed.

**Final Justification:**

I maintain my score.

**Key Questions For Authors:**

1. Please provide the details of training loss.
2. The algorithmic details of the interface-based flux and P0DG-based flux schemes are not provided. The current description does not clearly differentiate between them.

Minor remarks

1. In Figure 2, the caption indicates that there are three subfigures. However, only two subfigures are shown.
2. In the caption of Figure 3, ‘kernel’ should be capitalized.

**Strengths And Weaknesses:**

Strengths
1. The proposed Discontinuous Galerkin Neural Operator (DGNO) seems to align well with the spatially varying physics of defocus blurrring.
2. The proposed DGNO can be incorporated into several lifting modules, improving the recovery performance.
3. The experiments demonstrate the effectiveness of the method.

Weakness:
1. The training loss is not mentioned in the paper.
2. The details of interface-based flux and P0DG-based flux schemes are not well described.

---

> ### Author Rebuttal · Authors · 2026-03-31
>
> We thank the reviewer for the positive assessment and for recognizing that DGNO is well aligned with the spatially varying physics of defocus blur, can be flexibly incorporated into different lifting modules, and achieves strong empirical performance.  We will clarify these points in the revision.
>
> **Question 1: Training loss.**
> In our implementation, we use a spatial--frequency combined loss over three scales:
>
> $L_s=\sum_{s=1}^3\frac{1}{E_s}\|\hat Y_s-Y_s\|_1,$
>
> $L_f=\sum_{s=1}^3\frac{1}{E_s}\|\mathcal F(\hat Y_s)-\mathcal F(Y_s)\|_1,$
>
> $L=L_s+\lambda L_f,\;\lambda=0.1$
>
> where $s$ is the scale index, $\mathcal{F}$ denotes FFT, $E_s$ is a normalization factor, and $\hat{Y}_s$ and $Y_s$ are the restored and target images, respectively. The first term enforces spatial reconstruction fidelity, while the second promotes frequency-domain consistency. We will add this formulation explicitly in the revised manuscript.
>
> Reference:
> [a] Cho, Sung-Jin, et al. "Rethinking coarse-to-fine approach in single image deblurring." Proceedings of the IEEE/CVF international conference on computer vision. 2021.
>
> **Question 2: Clarifying interface-based flux vs. P0DG-based flux.**
>
> The algorithmic details of both schemes are provided in Section 3.2 (Interface-based Numerical Flux Operator and P0DG-based Numerical Flux Operator), with operator-level flux formulations given in Appendix C.
>
> Specifically, Eq. (14) defines the interface-based flux, where the interface operator is first constructed via Eq. (13). For a shared interface, the left element’s right-boundary operator and the right element’s left-boundary operator are coupled through a numerical flux (Appendix C, Eqs. 28, 29, 30, and 32). This coupling ensures that, during the volume integration of each element, information from its neighboring element is incorporated. As a result, all elements are weakly connected through interface fluxes, enabling global coupling while preserving locality.
>
> In contrast, the P0DG-based flux follows a similar coupling mechanism but replaces interface operators with cell-wise (volume) operators. That is, inter-element information exchange is performed directly through element-wise representations rather than explicit interface terms.
>
>
> **Minor presentation issues.**
>
> - Figure 2 caption mismatch: Figure 2 contains only two subfigures, (a) and (b), and the mention of panel (c) in the caption is an error.We will correct the caption as follows:Figure 2. Overview of defocus blur formation and operator-learning approaches. (a) Defocus-blur formation. (b) SRNO versus the proposed DGNO.
> - Figure 3 caption capitalization: we will fix the capitalization of “Kernel” for consistency.

---

> > ### Author Rebuttal · Reviewer_1z9U · 2026-04-03
> >
> > Thank you for the response. However, it remains unclear whether the improved performance stems from the proposed method itself or from the use of a different training loss, particularly since a three-scale loss is adopted. To ensure a more comprehensive and convincing evaluation of DGNO, the authors should provide fair comparisons and additional ablation studies. For example, experiments using a single-scale loss, as well as applying the same training loss to competing methods, would help better isolate the contribution of DGNO.

---

> > > ### Author Response · Authors · 2026-04-06
> > >
> > > Thank you for the insightful comment. We would like to clarify that in Table 2, both GG and DG are trained using the same three-scale loss, and the observed improvement from GG to DG already demonstrates the effectiveness of the proposed DG design itself. To further provide a more comprehensive and convincing evaluation of DGNO, we conducted additional ablation studies. Specifically, DGNO achieves 36.80 dB under a single-scale loss, which still outperforms MPT (35.44 dB) by 1.36 dB, indicating that the performance gain mainly comes from the proposed method rather than the loss design. Moreover, when applying the same three-scale loss to MPT, its performance slightly improves to 35.62 dB, but still remains significantly lower than DGNO (37.22 dB), with a gap of 1.60 dB. These results further confirm that the superiority of DGNO primarily stems from the proposed architecture rather than the use of a different training loss.
> > >
> > > We appreciate your review and feedback. In addition to the new experiments on the single-scale and three-scale loss, we hope our responses address all of your concerns in the review. Please let us know if you have any further questions after reading our rebuttal. We aim to address all potential issues during this discussion period and hope you will consider raising the score.

---

### Official Review · Reviewer_PXDk · 2026-03-12

**Soundness:** 2
**Presentation:** 2
**Significance:** 2
**Originality:** 3
**Overall Recommendation:** 4
**Confidence:** 3

**Summary:**

This paper studies pathology defocus deblurring through the lens of operator learning. Rather than treating deblurring as a standard finite-dimensional image-to-image regression problem, the paper argues that pathological defocus blur is better modeled as the inversion of a spatially varying integral operator. Based on this view, the authors propose a Discontinuous Galerkin Neural Operator (DGNO), which decomposes the operator into element-local volume operators and interface numerical fluxes inspired by discontinuous Galerkin methods. Two variants are introduced, one using face-based fluxes and one using a lighter P0DG-style cell formulation. Experiments on pathology microscopy datasets show improvements over several image restoration baselines, and the paper also includes ablations on flux design, boundary conditions, real-world transfer, and downstream cell detection.

**Compliance With Llm Reviewing Policy:**

Affirmed.

**Final Justification:**

This paper studies pathology defocus deblurring from an operator-learning perspective and proposes a Discontinuous Galerkin Neural Operator (DGNO) that combines element-local operators with interface-based coupling. I continue to find the problem relevant and the core architectural idea reasonably original. In particular, the decomposition into local volume operators and interface numerical fluxes is a thoughtful adaptation of DG-style ideas to neural operators, and the empirical results on the pathology datasets are strong.

My original concerns were mainly about soundness, scope of significance, and conceptual clarity. In the rebuttal, the authors addressed these concerns in a meaningful way. The added controlled synthetic experiments directly strengthen the central claim that DGNO is better suited than global Galerkin operators for spatially heterogeneous blur, and the additional boundary-region results further support this point. The sensitivity study over element size and operator iterations also helps show that the method is not overly fragile to these design choices. In addition, the new RealDOF results, together with the existing DPDD evaluation, make the transfer discussion more convincing than in the original submission. The clarification of what is meant by “DG-inspired” versus a literal inheritance of classical DG theory also resolves an important conceptual ambiguity. Finally, the added runtime, memory, and scaling results make the practical trade-offs clearer.

Overall, I still view the contribution as primarily an architectural and application-driven advance rather than a foundational machine learning contribution, and some limitations in generality remain. However, after considering both the paper and the rebuttal, I believe the main concerns from my original review have been addressed sufficiently well. On balance, I now view the paper more positively than I did initially, and my final recommendation is correspondingly more favorable.

**Key Questions For Authors:**

1. The paper argues that DGNO is better suited than global Galerkin operators for spatially heterogeneous and locally discontinuous blur. Can the authors provide a more formal justification for this claim, either theoretically or through a more controlled synthetic experiment with known spatially varying kernels?

2. How sensitive is performance to the element size and the number of operator iterations T? These seem to be important modeling choices, but the current paper does not fully characterize their effect.

3. What aspects of DGNO transfer beyond pathology deblurring? Can the authors provide evidence, even limited, that the proposed decomposition is useful for other spatially heterogeneous inverse problems or other non-pathology datasets?

4. The appendix explains that DGNO is not derived from a differential formulation and that the interface terms arise from operator discretization rather than classical integration by parts. Can the authors clarify more explicitly in the main paper what is meant by 'DG-inspired' versus what properties are actually inherited from classical DG methods?

5. The paper would benefit from a clearer discussion of computational trade-offs, including runtime, memory usage, and scaling behavior as image size or partition granularity increases. This would help readers assess practicality.

**Limitations:**

No. The paper should more explicitly discuss at least three limitations: (1) the current evidence is concentrated on pathology microscopy and does not yet establish broad applicability; (2) the DG interpretation is architectural rather than a faithful transfer of classical DG theory; and (3) the method may depend on specific discretization choices such as element size and interface design. A short discussion of these limits in the paper would make the claims more balanced.

**Strengths And Weaknesses:**

This submission has several strengths. First, the problem is meaningful: pathology defocus deblurring is practically relevant, and the paper gives a plausible motivation for why spatially varying blur may be better viewed at the operator level rather than through standard shift-invariant convolutional assumptions.

Second, the proposed DGNO architecture is reasonably original at the design level. The decomposition into element-local operators plus interface coupling is a thoughtful adaptation of discontinuous Galerkin ideas to neural operators, and the paper does more than simply rename an existing attention block.

Third, the empirical results on the target pathology datasets are competitive and often strong, especially on BBBC006w1 and BBBC006w2, where DGNO improves over several well-known restoration baselines. The inclusion of analyses on local/global operators, numerical fluxes, boundary conditions, and downstream detection is also appreciated.

My main concerns are about soundness, presentation, and machine learning significance. On soundness, the paper’s central motivation is appealing, but many of the stronger claims are supported mostly by intuition and empirical evidence rather than rigorous analysis. In particular, the paper argues that DGNO is better suited for spatially heterogeneous and locally discontinuous blur, yet it does not provide a convincing formal account of why this decomposition should yield better approximation, stability, or generalization properties than global Galerkin-style operators.

There is also some conceptual looseness in the DG analogy: the appendix notes that DGNO is not derived from a differential formulation and that the interface terms arise from discretized operator decomposition rather than true integration-by-parts structure. That makes the DG interpretation interesting, but also less principled than the paper sometimes suggests.

On presentation, the paper is readable at a high level, but the writing is uneven and occasionally imprecise. Some figure captions are awkward or unclear, notation is not always introduced cleanly, and several claims sound stronger than the evidence supports.

On significance, the work is relevant but currently too narrow for ICML. The strongest evidence is concentrated on pathology microscopy deblurring, which supports the application claim but not yet a broader machine learning contribution. The results outside the pathology setting appear limited, so the paper does not yet convincingly show that DGNO is a generally useful neural-operator design principle beyond this domain.

On originality, I view the paper somewhat more positively. The operator-learning perspective on pathology defocus deblurring is interesting, and the DG-style decomposition is a creative architectural idea. Still, the paper’s novelty is more architectural and domain-specific than foundational.

Overall, I think the paper has clear merit and could become a stronger submission after revision, but in its current form the weaknesses outweigh the strengths for ICML.

---

> ### Author Rebuttal · Authors · 2026-03-31
>
> We sincerely thank the reviewer for the careful reading and thoughtful feedback. We address each concern in detail below and will revise the paper accordingly.
>
> **Question 1: Why DGNO for Spatially Heterogeneous Blur?**
>
> To more directly test this point, we added two controlled synthetic experiments. First, on 50 randomly generated synthetic images, we constructed sharp images composed of squares, hollow boxes, and thin lines, and applied known spatially varying Gaussian blur, where the blur strength at each pixel was controlled by a predefined sigma map. This setting allows us to isolate the restoration ability under spatially heterogeneous blur with fully known degradation kernels, without interference from real-data distribution bias. The results show that DGNO consistently outperforms the global Galerkin baseline under different blur ranges: when the sigma range is 8–10, the average PSNR of GG, DGNO-Face, and DGNO-Cell is 34.69, 38.96, and 37.91 dB, respectively; when the range is expanded to 0.6–13, the corresponding values are 44.24, 46.23, and 46.32 dB. The visual results (https://anonymous.4open.science/r/visual1-04B1/rebullt_synthetic%20experiment.pdf) are also consistent with this finding: around thin lines, hollow-box boundaries, and other local high-frequency structures, GG tends to leave more residual blur or lose local structures, whereas DGNO-Cell recovers thin edges and local contours more faithfully and exhibits weaker residuals. Second, on BBBC006$_{w1}$ (153 images), We further constructed synthetic spatially varying blur using known Gaussian kernels under a random spatial variation setting, with the sigma range set to 0.6–13. Under this setting, DG improves PSNR over GG by 1.94 dB (25.81→27.75 dB), and still achieves a 1.77 dB gain on boundary regions, indicating its stronger ability to handle spatial transition areas.
>
> **Question 2: Sensitivity to element size and operator iterations**
>
> We ablate DGNO(cell) over element size (Win) and iterations T on BBBC006w1. It is insensitive: for Win=4–16 and T=1–4, PSNR stays 37 dB with <0.3 dB variation. Best is Win=8, T=2; degradation appears only at Win=32, indicating robustness.
>
> |Win\T|1|2|3|4|
> |-|-|-|-|-|
> |4|36.94|36.86|37.11|36.89
> |8|36.89|37.22|37.10|37.04
> |16|37.07|37.11|37.01|36.98
> |32|36.85|36.91|36.41|36.39
>
> Table A. Sensitivity of DGNO(cell) to element size and operator iteration number T.
>
> **Question 3: What transfers beyond pathology deblurring?**
>
> We already evaluated the method on the non-pathology DPDD dataset in Table 4. To further address the question of transfer beyond pathology, we additionally conducted experiments on the RealDOF test set. As shown in the table B, DGNO again achieves the best performance.
>
> |Method|PSNR|SSIM|LPIPS|
> |-|-|-|-|
> |DPDNet|22.52|0.644|0.583
> |IFAN|24.71|0.748|0.304
> |EAMamba|24.59|0.761|0.309
> |Ours|25.08|0.781|0.275
>
> Table B: Quantitative evaluation on the RealDOF test set.
>
> **Question 4:  Clarification on the DG inspiration.**
>
> DGNO is built upon the integral formulation (2) of neural operators, where each layer is defined as a kernel integral operator over the domain. Such operators do not involve differential terms and hence do not admit a natural integration-by-parts procedure. As a result, interface terms do not arise at the continuous operator level.
>
> By “DG-inspired,” we mean an operator-level decomposition analogous to the one induced by integration by parts. Specifically, DGNO decomposes the global integral operator into element-wise local operators and interface coupling operators. The local integral part is already justified in SRNO, while the interface part is modeled at the operator level using classical DG-style numerical fluxes. Building on this, we further introduce both face-wise flux modeling and a P0DG variant, providing an operator-level way to realize discontinuous Galerkin ideas within neural operators.
>
> We will revise the main paper to clarify this distinction.
>
>
> **Question 5: Computational trade-offs: runtime, memory, scaling**
>
> As illustrated in Fig. 1, DGNO already shows strong efficiency from both the performance–parameter and resolution–FLOPs perspectives. To further quantify this advantage, we add Table C reporting GPU memory and throughput across image sizes, which confirms its superior runtime–memory trade-off and scalability.
>
> |Size|SwinIR|Restormer|MPT|DGNO|
> |-|-|-|-|-|
> |128|4/29|3/24|2/7|1/30
> |256|16/5|12/11|10/9|4/27
> |384|36/2|27/5|24/3|9/18
> |512|63/1|48/3|41/2|15/10
>
> Table C: Memory (GB) / Throughput (img/s) vs. image size
>
> **On significance**
> We disagree that the scope is narrow: spatially-varying deblurring is a broadly relevant problem across domains (e.g., pathology, microscopy, astronomy), and as noted by reviewer YYoJ, it is “a real and important issue,” while our additional experiments (e.g., DPDD and RealDOF) further demonstrate the generality of the proposed framework.

---

> > ### Author Rebuttal · Reviewer_PXDk · 2026-04-03
> >
> > Thank you for the detailed rebuttal. My main concerns have been adequately addressed. The added controlled synthetic experiments directly strengthen the central empirical claim that DGNO is better suited than global Galerkin operators for spatially heterogeneous blur, and the new boundary-region results further support this point. The added sensitivity study over element size and operator iterations also helps clarify that the method is not overly fragile to these design choices. In addition, the new RealDOF results, together with the existing DPDD evaluation, make the transfer discussion more convincing than in the original submission. The clarification of what is meant by “DG-inspired” versus a literal inheritance of classical DG theory also resolves an important conceptual ambiguity that I had raised. Finally, the added runtime, memory, and scaling results make the practical trade-offs clearer. Overall, the rebuttal addresses the core concerns in my original review sufficiently well, and I would revise my overall assessment upward.

---

> > > ### Author Response · Authors · 2026-04-06
> > >
> > > Thank you very much for your positive and detailed feedback. We are glad that the additional experiments and clarifications have addressed your concerns and strengthened both the empirical validation and conceptual clarity of the work. We will incorporate all these revisions into the final manuscript to further improve the clarity of the contributions and the comprehensiveness of the evaluation.

---

### Official Review · Reviewer_GGkq · 2026-03-13

**Soundness:** 2
**Presentation:** 3
**Significance:** 2
**Originality:** 3
**Overall Recommendation:** 4
**Confidence:** 4

**Summary:**

The paper proposes a discontinuous Galerkin neural operator that decomposes a learned operator into element-local volume operators and interface numerical fluxes to address the defocus deblurring under spatially varying and locally discontinuous blur, commonly arised in pathology imaging.

**Compliance With Llm Reviewing Policy:**

Affirmed.

**Final Justification:**

Having reviewed the author's rebuttal and other reviewers' comments, I have slightly increased my score as some of my concerns were resolved. However, I remain somewhat on the fence about the paper's overall merit, especially its practicality. I will defer to the AC for the final decision.

**Key Questions For Authors:**

I do not have any separate questions. My primary inquires for the paper are directly reflected in the weakness section.

**Limitations:**

no potential negative societal impact

**Strengths And Weaknesses:**

+ The paper provides clear formulation on their use of Galerkin neural operator  on the clearly defined problem and architecture contribution.
+ The paper has a wide range of baseline experiments and ablation analysis
+ The paper addresses generalization capability of the proposed method.

However, I see the weakness of the present work as follows:

-  somewhat unclear qualitative improvements compared to the baselines. For example, I see Restormer and MPT+EFCR results are slightly better than the proposed one (Figure 4).
- Unclear implication of the interpretability (Figure 7):  it may provide qualitative intuition that DGNO may rely more on local boundary information. The fact that the maps look more localized or visually appealing is not sufficient evidence that the learned operator is more physically grounded. I would not see it is a strong validation of the proposed method.
- While global PSNR and SSIM is effective to evaluate overall reconstruction quality, they may not reveal the degradation of important structures of the biological samples since the proposed method targets for pathology applications. Evaluating ROI-specific metrics, such as MSE or SSIM would be a more stronger evidence for the practicality of the present approach.

---

> ### Author Rebuttal · Authors · 2026-03-31
>
> We sincerely thank the reviewer for the careful reading and constructive feedback. We address the main concerns below.
>
> **Weaknesses 1: On the qualitative comparison in Fig. 4**
>
> To make the visual comparison clearer, we added both arrow annotations and residual visualizations in the revised figure; the updated visual results can be found in the linked figure(https://anonymous.4open.science/r/visual-5CC3/Visual%20comparison.pdf). These additions make the differences in boundary sharpness, local structure recovery, and remaining blur artifacts more explicit. The qualitative observations are also consistent with the region-specific quantitative results: on BBBCw1, Cell/Face achieve 26.17/26.33 dB, compared with 22.64 dB for Restormer and 22.35 dB for MPT; on BBBCw2, Cell/Face achieve 21.17/21.37 dB, compared with 20.07 dB and 19.49 dB; on 3DHistech, Cell/Face achieve 31.64/31.53 dB, compared with 31.12 dB and 26.22 dB. These results support that the visual improvements are consistent with stronger restoration in biologically relevant regions.
>
>
> **Weaknesses 2: On the implication of interpretability in Fig. 7**
>
> We did not intend to claim that DGNO simply “relies more on local boundary information.” The red box in the sharp image marks the queried region, and Fig. 7 visualizes its corresponding LAM response. The key point is that DGNO, although built from element-local modeling, still produces a global response: the activation is not confined to the red box (Sharp) itself, but extends to other spatially correlated cellular structures across the image. At the same time, compared with NAFNet, MPT, and especially SRNO, DGNO avoids many irrelevant diffuse responses in unrelated regions (see the blue arrows), whereas SRNO shows stronger undesired spread and interference (red arrow). Therefore, Fig. 7 is intended to show that DGNO achieves global interaction with controlled, task-relevant responses through interface fluxes, rather than merely producing more localized or visually appealing maps.
>
> **Weaknesses 3: On pathology-relevant evaluation beyond global PSNR/SSIM**
> Fig.6 is already region-specific rather than limited to global image-level metrics. In the original manuscript, we separately analyzed biologically meaningful regions and reported regional PSNR for boundary-relevant and non-boundary areas, showing that DG achieves higher reconstruction quality in both regions. Specifically, DG improves PSNR from 31.72 dB to 32.22 dB in the edge band, and from 43.07 dB to 43.84 dB in the non-edge region. This already indicates that the benefit of DG is not merely a global average effect, but also appears in spatially localized, biologically relevant regions.
>
> To further strengthen this point, we have now supplemented Fig.6 with regional SSIM and MSE on the same response regions. In the edge band, DG improves SSIM from 0.925 to 0.928 and reduces MSE from 6.7e-4 to 6.0e-4. In the non-edge region, DG maintains SSIM at 0.911 while further reducing MSE from 4.93e-5 to 4.13e-5. These additional metrics provide complementary evidence that DG better preserves local structural fidelity, especially around cellular boundaries that are more relevant for biological interpretation. Moreover, the downstream results in Table 7 directly evaluate whether the restored images better support subsequent biological analysis, where DGNO achieves the best cell detection performance among restoration baselines, further confirming its practical value beyond pixel-level restoration.

---

> > ### Author Rebuttal · Reviewer_GGkq · 2026-04-05
> >
> > Thanks for the clarifications and additional explanations.

---

> > > ### Author Response · Authors · 2026-04-06
> > >
> > > Thank you very much for your positive feedback and for acknowledging that your concerns have been fully addressed. We are glad that the additional clarifications and experiments helped improve the quality and completeness of the paper. We will incorporate all these revisions into the final manuscript to further improve the clarity of the contributions and the comprehensiveness of the evaluation. We would greatly appreciate it if you could kindly consider updating your score to reflect your final assessment.

---

### Decision · Program_Chairs · 2026-04-30

**Decision:**

Accept (regular)

**Comment:**

The authors present a defocus deblurring method, DGNO.

All reviewers acknowledge the originality of the proposed method while raising concerns in the first round of review. The authors provided the rebuttals addressing the concerns. Most of the reviewers' concerns were resolved from the discussion with the authors.

All reviewers suggested weak accept as their final score.
As the consensus is reached, I recommend acceptance.